# Enhancing in vivo cell and tissue targeting by modulation of polymer nanoparticles and macrophage decoys

Alexandra S. Piotrowski-Daspit [1,2,3,11] ✉, Laura G. Bracaglia[1,4,11] ✉, David A. Eaton [1], Owen Richfield[1], Thomas C. Binns [1,5], Claire Albert[1], Jared Gould[1], Ryland D. Mortlock [1], Marie E. Egan[6,7], Jordan S. Pober [7,8] & W. Mark Saltzman [1,7,9,10] ✉

The in vivo efficacy of polymeric nanoparticles (NPs) is dependent on their pharmacokinetics, including time in circulation and tissue tropism. Here we explore the structure-function relationships guiding physiological fate of a library of poly(amine-co-ester) (PACE) NPs with different compositions and surface properties. We find that circulation half-life as well as tissue and cell-type tropism is dependent on polymer chemistry, vehicle characteristics, dosing, and strategic co-administration of distribution modifiers, suggesting that physiological fate can be optimized by adjusting these parameters. Our high-throughput quantitative microscopy-based platform to measure the concentration of nanomedicines in the blood combined with detailed biodistribution assessments and pharmacokinetic modeling provides valuable insight into the dynamic in vivo behavior of these polymer NPs. Our results suggest that PACE NPs—and perhaps other NPs—can be designed with tunable properties to achieve desired tissue tropism for the in vivo delivery of nucleic acid therapeutics. These findings can guide the rational design of more effective nucleic acid delivery vehicles for in vivo applications.

Targeted drug delivery in vivo is a complex challenge, with multiple barriers at the levels of the organism, organs/tissues, and individual cells[1]. Non-viral lipid- or polymer-based vehicles can improve the delivery efficiency of therapeutic molecules. In the context of nucleic acid delivery, while non-viral vehicles can be effective at cargo encapsulation, protection of cargo from nuclease degradation, and transport across the cell membrane, the primary challenge is still sufficient accumulation in target tissues instead of non-target tissues, where they can lead to side-effects. Standard therapeutic

development pipelines, in which candidate vehicles are identified in cell culture studies, are often not predictive of what happens when these agents are administered to an animal. For example, we and others have found differences in vehicle characteristics affecting cell uptake and release of cargo following in vitro vs. in vivo delivery[2–4]. There are good reasons for this: traditional cell culture models are simplified and often contain one cell type without normal tissue architecture, fluid flows, and other complexities found in vivo. Several studies have highlighted the need to study delivery in more

[1]Department of Biomedical Engineering, Yale University, New Haven, CT, US. [2]Department of Biomedical Engineering, University of Michigan, Ann Arbor, MI, US. [3]Department of Internal Medicine – Pulmonary and Critical Care Medicine Division, Michigan Medicine, University of Michigan, Ann Arbor, MI, US. [4]Department of Chemical and Biological Engineering, Villanova University, Villanova, PA, US. [5]Department of Laboratory Medicine, Yale School of Medicine, New Haven, CT, US. [6]Department of Pediatrics, Yale School of Medicine, New Haven, CT, US. [7]Department of Cellular & Molecular Physiology, Yale School of Medicine, New Haven, CT, US. [8]Department of Immunobiology, Yale School of Medicine, New Haven, CT, US. [9]Department of Dermatology, Yale School of Medicine, New Haven, CT, US. [10]Department of Chemical & Environmental Engineering, Yale University, New Haven, CT, US. [11]These authors contributed equally: Alexandra S. Piotrowski-Daspit, Laura G. Bracaglia. ✉e-mail: asapd@umich.edu; laura.bracaglia@villanova.edu; mark.saltzman@yale.edu

physiologically relevant contexts, including high-throughput screens in vivo[5].

The majority of studies on polymeric vehicles encapsulating nucleic acids employ local delivery unless hepatic tissue is the disease target[1]. Local or compartmental administration methods—such as intranasal or intratracheal instillation to the lungs or intraocular administration—can circumvent systemic clearance mechanisms. However, local delivery is not suitable for the treatment of in situ tumors, internal organs, and diseases that affect multiple organ systems. Moreover, it may not provide access to the disease-associated cell types within a tissue. Intravenous (IV) administration can theoretically provide opportunities for delivery vehicles to reach almost any tissue in the body. But there are barriers; the single most formidable barrier to IV delivery of particulate carriers is clearance by the mononuclear phagocyte system (MPS) (sometimes also known as the reticuloendothelial system (RES))— i.e., vehicle uptake by intravascular phagocytic cells present primarily in the liver and spleen[6-9]. Clearance by these phagocytes reduces the dose of vehicles available in circulation and limits possible accumulation in intended tissues. To overcome this barrier, vehicles can be designed with a strong affinity to a particular tissue (tissue tropism) so that they can accumulate in that tissue faster than they are cleared by phagocytes. Other carriers with abilities to evade phagocyte clearance or extend circulation time allow for more chances to accumulate in areas of interest. Understanding the characteristics that define the behavior of delivery vehicles in vivo is important for future advances in delivery vehicle design.

In this work, we use a library of polymeric delivery vehicles as well as high-throughput tools to study the structure-function relationships guiding the physiological fate of nanomedicines. Poly(amine-co-esters) (PACEs) are a family of tunable, biodegradable, and biocompatible polymers designed for nucleic acid delivery (Fig. 1). These materials can be used to deliver a wide range of nucleic acids, such as siRNA, mRNA, and pDNA, both in cell culture and in vivo[10-12]. The ability to control the physical and chemical properties of PACEs—as well as the size and surface properties of the resulting vehicles including the incorporation of targeting ligands—makes this platform ideal for studying the effects of polymer chemistry and nanoparticle (NP) characteristics on physiological fate in vivo after IV delivery. We also use PACE vehicles to determine the effects of dosing[6] and strategic co-administration of multiple formulations on NP clearance rate. To assess these variables, we employ a recently developed high-throughput tool to measure NP blood concentration[13] combined with thorough biodistribution assessment using whole organ imaging and flow cytometry. We then use these comprehensive data sets to train a pharmacokinetic model describing PACE NP circulation and biodistribution. This combined approach—using multiple experimental measurements with physiology-based computer models—enables us to gain a comprehensive understanding of delivery vehicle fate, and to determine the factors that influence delivery. Overall, we identify several parameters that are key for controlling the biodistribution of polymeric NPs. NP properties such as size, charge, and morphology play an important role but are often constrained by the type of cargo one intends to deliver. Our multimodal experiments and computational modeling establish that dosing, co-administration of decoys to limit phagocytic clearance, and antibody-mediated targeting can impact the physiological fate and therapeutic effects of systemic NP delivery.

## Results

### Polymer chemistry and NP characteristics define blood concentration and biodistribution

We first studied the effects of polymer chemistry and NP characteristics on in vivo delivery. Taking advantage of our PACE library of materials (Fig. 1a), we prepared NP formulations with various PACE polymer chemistries: PACE (an unmodified PACE polymer), PACE-

COOH, and PACE-PEG. We chose these materials as they have been previously well-characterized with demonstrated efficient nucleic acid loading and delivery[10]. From these polymers, we prepared DiD-loaded NP batches, ranging in diameter from 140 nm to 240 nm (Supplementary Table 1). We administered the PACE NPs to mice intravenously (at a final blood concentration of 250 µg/mL). Blood concentration (Fig. 1b) over time was measured from each animal using our high-throughput quantitative microscopy-based method, which requires less than 1% of the blood volume of the animal[13]. A comprehensive study of tissue and cell-type tropism was performed using whole organ IVIS imaging, and flow cytometry of homogenized heart, lung, liver, spleen, kidney, and bone marrow (Fig. 1c–g). Based on our previous studies using poly(lactic-co-glycolic acid) (PLGA) NPs, we have not observed dye leakage from solid polymeric NPs within the 48- h time period during which blood concentration and biodistribution is assessed[13], suggesting that fluorescent signal from the dye is a reliable indicator of NP location. As expected, PACE-PEG NPs exhibited prolonged blood circulation compared to unPEGylated NPs (Fig. 1b) as well as widespread distribution (Fig. 1c-I, Supplementary Fig. 1). The smallest PACE-PEG NPs exhibited the broadest biodistribution and highest levels of uptake (Fig. 1), consistent with the hypothesis that smaller NPs enable more widespread tissue distribution[14]. Interestingly, the same relationship was not true for the other PACE chemistries: PACE and PACE-COOH. For unmodified PACE, the largest NPs trafficked only to the liver and spleen; the medium PACE NPs exhibited the broadest distribution, with NPs present also in the heart, lung, and bone marrow; the smallest PACE NPs exhibited broader distribution than the largest but were not as widespread as the medium PACE NPs. In contrast, the largest PACE-COOH NPs exhibited the most widespread distribution within this group, with biodistribution directly decreasing with size. The relationship between NP size, chemistry, circulation half-life, and general distribution beyond the liver is illustrated in Fig. 1h, i. As shown, no clear trend is apparent. These heterogeneous results suggest that both polymer chemistry and nanoparticle characteristics play a significant role in determining physiological fate, and that size alone cannot accurately predict NP behavior in vivo. These results for polymer NPs are consistent with recently discovered characteristics such as composition, charge, and size affecting lipid NP tissue tropism[15]. Beyond NP characteristics, we have also found that disease state/pathophysiology can impact biodistribution. We compared the biodistribution of a subset of PACE NPs in a cystic fibrosis (CF) mouse model harboring the F508del CF mutation[16], and while the general trends are the same in that PACE-PEG NPs accumulate more broadly than unPEGylated NPs, PACE NPs also accumulated in the lungs of CF animals in contrast to our observations in wild-type animals, where no lung accumulation was observed (Supplementary Fig. 2).

### Dosing affects circulation half-life and biodistribution

A recent study demonstrated that the dose of PEGylated gold NPs affects the NP circulation half-life due to rate-limited phagocyte-mediated clearance[6]. To understand if the same phenomena can affect polymer NPs, we varied the dose of administered dye-loaded PACE-PEG NPs between 0.1 mg and 3.5 mg per animal (Fig. 2). Using estimates of NP size and density (180 nm and 1 g/mL, respectively) these doses cover a range of 0.03 trillion to 1 trillion NPs per mouse. As the NP dose increased, the circulation half-life of the NPs also increased, from 23 hr to 90 hr (Fig. 2a, b). As expected, NPs were found in abundance in the liver and spleen at all doses, with increasing fluorescence detected in each organ using IVIS imaging as the dose increased (Fig. 2c, d). This effect was also observed in single-cell analysis by flow cytometry, both in terms of NPs per cell (mean fluorescence intensity (MFI) per cell) and in terms of the percent of cells with a certain threshold of fluorescently labeled NPs above background (Fig. 2e–h).

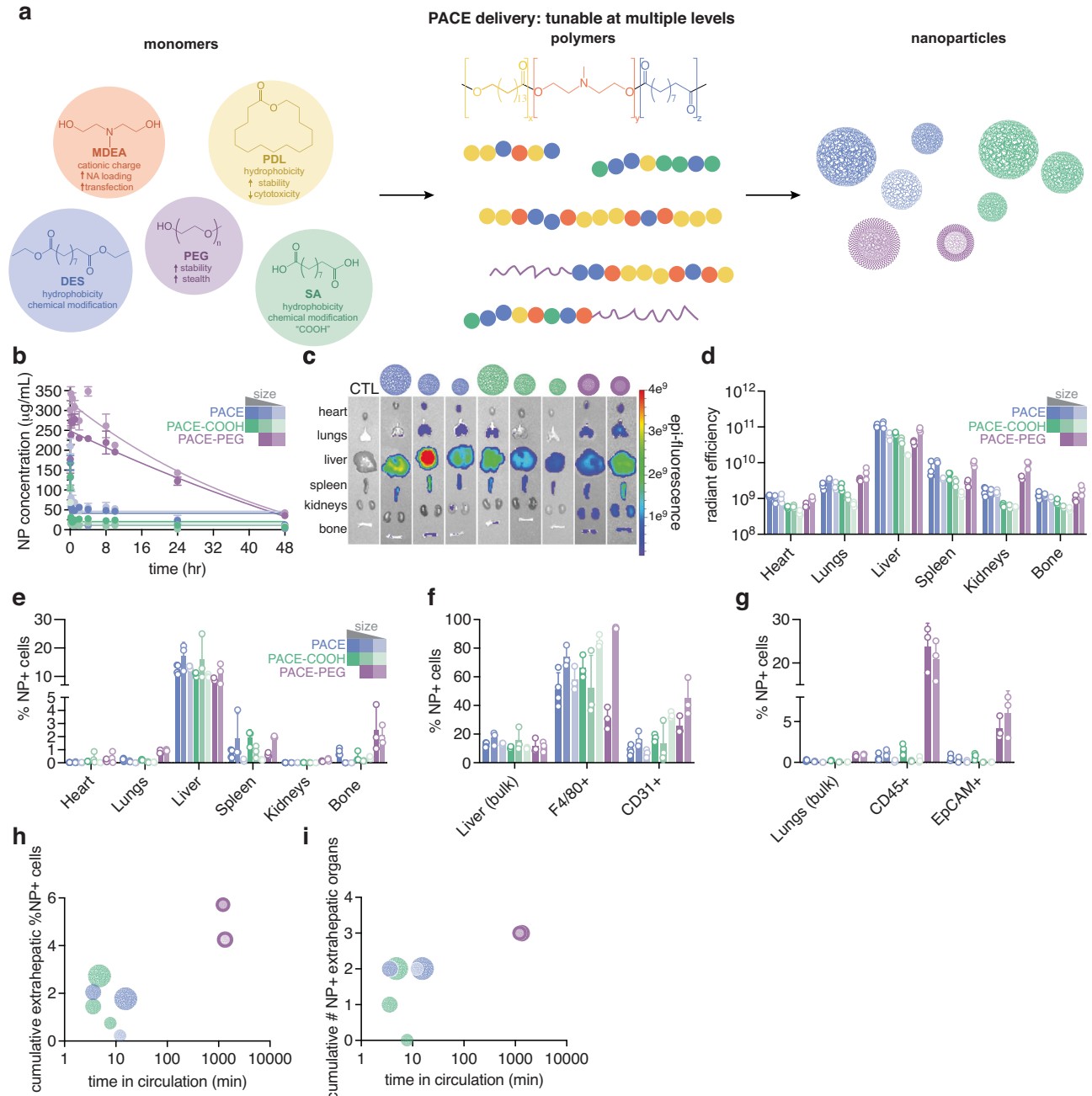

**Fig. 1 | Effects of PACE composition and NP characteristics on blood concentration and biodistribution. a** Schematic of tunable PACE family polymers from monomer building blocks to formulated NPs, NA loading: nucleic acid encapsulation into NPs. **b** Blood NP concentration over time following systemic IV injection of dye-loaded PACE (blue), PACE-COOH (green), and PACE-PEG NPs (purple) of various sizes (see Supplementary Table 1) using quantitative microscopy (*n* = 3 mice per group per time point; error bars represent standard deviation (SD)). Representative end-point (**c**) IVIS analysis of PACE NP uptake in various organs (heart, lungs, liver, spleen, kidneys, and bone), CTL: untreated control. End-point analyses of (**d**) whole organ fluorescence quantification of PACE NP uptake in various organs (*n* = 3 mice per group per organ; error bars represent SD), (**e**) %NP+ cells in homogenized organs by flow cytometry (*n* = 3 mice per group per organ; error bars represent SD), (**f**) %NP+ cells in homogenized liver populations (bulk, F4/80+, and CD31+) by flow cytometry (*n* = 3 mice per group per population; error bars represent SD), and (**g**) %NP+ cells in homogenized lung populations (bulk, CD45+, and EpCAM+) by flow cytometry (*n* = 3 mice per group per population; error bars represent SEM). The effects of PACE NP characteristics on biodistribution as measured by (**h**) cumulative extrahepatic %NP+ cells and (**i**) cumulative number of NP+ extrahepatic organs, both as a function of time in circulation. PACE NPs are represented according to chemistry (PACE (blue), PACE-COOH (green), and PACE-PEG NPs (purple)) and size (see Supplementary Table 1).

Interestingly, as the dose increased, the ratio of NPs in the liver versus other organs, such as the lungs or kidneys, decreased. This suggests phagocytic cells in the liver begin to reach capacity, allowing NPs to find other targets. Indeed, the single-cell analysis using flow cytometry showed a similar MFI in liver F4/80+ cells in the 3 largest doses (Fig. 2h), suggesting these cells may have reached a near

maximum NP quantity at the 2 mg dose. Concurrent with this potential saturation, the ratio of lung cells with NPs over liver cells with NPs increased from less than 1% at the 0.1 mg dose to 24% at the 3.5 mg dose (calculated from Fig. 2e). Similarly, the ratio of spleen cells with NPs over liver cells with NPs increased from 1% to 10% from 0.1 mg to 3.5 mg of NPs. These trends are summarized in Fig. 2k, l, which show

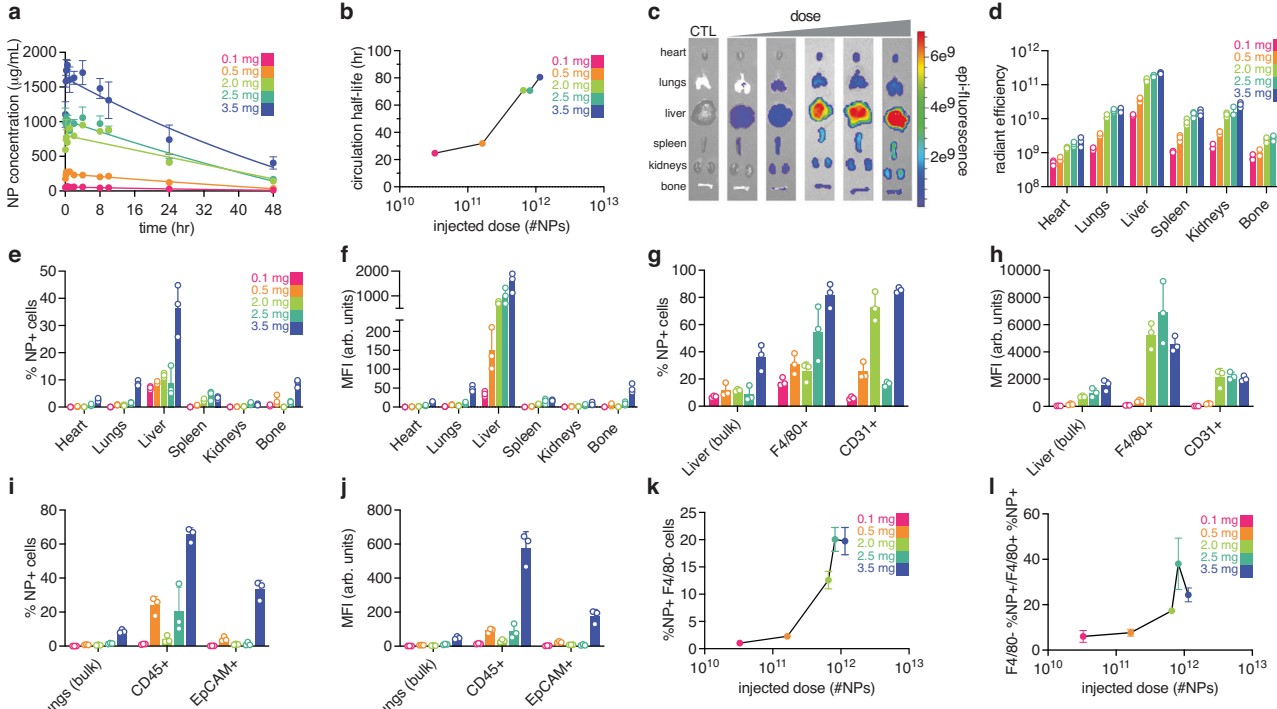

**Fig. 2 | Effects of PACE-PEG dosing on blood concentration and biodistribution.**
**a** Blood NP concentration over time following systemic IV injection of dye-loaded PACE-PEG NPs at various doses (0.1 mg: pink, 0.5 mg: orange, 2.0 mg: green, 2.5 mg: turquoise, and 3.5 mg: indigo per animal corresponding to approximately 30 billion, 160 billion, 650 billion, 800 billion, and 1 trillion NPs, respectively) using quantitative microscopy ($n = 3$ mice per group per time point; error bars represent standard deviation (SD)). **b** Circulation half-life of PACE-PEG NPs as a function of dose. Representative end-point (**c**) IVIS analysis of PACE-PEG NP uptake in various organs (heart, lungs, liver, spleen, kidneys, and bone), CTL: untreated control. End-point analysis of (**d**) whole organ fluorescence quantification of PACE-PEG NP uptake in various organs ($n = 3$ mice per group per organ; error bars represent SD).

End-point analyses of (**e**) %NP+ cells and (**f**) Mean fluorescence intensity (MFI) in arbitrary units (arb. units) of homogenized organs by flow cytometry ($n = 3$ mice per group per organ; error bars represent SD). End-point analysis of (**g**) %NP+ cells and (**h**) MFI in homogenized liver populations (bulk, F4/80$^+$, and CD31$^+$) by flow cytometry ($n = 3$ mice per group per population; error bars represent SD). End-point analyses of (**i**) %NP+ cells and (**j**) MFI in homogenized lung populations (bulk, CD45$^+$, and EpCAM$^+$) by flow cytometry ($n = 3$ mice per group per population; error bars represent SD). **k** End-point analyses of %NP + F4/80$^−$ cells in the liver as a function of dose ($n = 3$ mice per dose; error bars represent SD). **l** End-point analysis of the ratio of %NP + F4/80$^−$ cells to %NP + F4/80$^+$ cells in the liver as a function of dose ($n = 3$ mice per dose; error bars represent SD).

the percentage of non-F4/80$^+$ cells in the liver with NPs and the ratio of all non-F4/80$^+$ cells with NPs to F4/80$^+$ cells with NPs, respectively, both as a function of dose.

In addition to the percentage of cells with NPs, the quantity of NPs per cell was altered as well with increasing NP dose. The MFI of cells in the liver increased with increasing NP dose. However, looking specifically at the 3 highest doses, the liver MFI increased by a factor of 2 (Fig. 2f), whereas the lung (10x), bone marrow (30x) and kidney (4x) increased their MFI by a higher factor. Within the liver, this change was observed in the MFI of CD31$^+$ cells, which increased by a factor of 12 compared with bulk liver cells and the F4/80$^+$ cells, which each increased by about 5x. Overall, these patterns suggest that with increasing NP dose, RES cells in the liver begin to reach capacity, allowing NPs to accumulate in other tissues.

**PACE NP PBPK Modeling**
We developed and parameterized a Physiologically Based Pharmacokinetic (PBPK) model of PACE-PEG NPs in mice. A schematic of the model is shown in Fig. 3. We used this model to predict the concentration of NPs in the blood and organs at the experimentally tested doses (Fig. 4). The pharmacokinetics of the NPs are characterized by a gradual decline in the blood concentration, coinciding with accumulation of NPs in the liver. Other organ compartments such as the spleen and bone showed a swift initial uptake of NPs and then a gradual decline as NPs re-entered circulation from the tissue subcompartment. The heart and other tissues/brain compartments showed many orders

of magnitude lower NP uptake than the other organs, due to extremely low permeabilities (PA, Supplementary Table 3).

We used Monte Carlo Importance Sampling (MCIS) to reparameterize a PBPK model originally used to simulate the pharmacokinetics and biodistribution of PEGylated gold NPs[17]. The original parameter values corresponding to the PEGylated gold NPs as well as the new parameter values (represented as mean and standard deviations of gamma distributions) are tabulated in Supplementary Table 3. In comparing the parameter values for gold and PACE-PEG NPs, it is apparent that PACE-PEG NPs show substantially lower tissue uptake in comparison to gold NPs. For example, the liver's maximum phagocytosis rate is four orders of magnitude lower for PACE-PEG NPs than gold NPs. All organs other than the liver show a reduced uptake, both in vascular diffusion (PA) and phagocytosis ($K_{max}$), for PACE-PEG as opposed to gold NPs. Unlike in the case of PEGylated gold NPs, phagocytic PACE-PEG NP release parameters ($K_{rel}$) are higher than their corresponding maximum phagocytosis rate (with the exception of the liver), which reduces the amount of PACE-PEG NPs taken up by non-liver organs. Lastly, the kinetics of phagocytic uptake between gold and PACE-PEG NPs is vastly different; as opposed to a $K_{50}$ of 24 h for gold NPs in all organs, the liver and spleen both show an almost immediate progression to the maximum phagocytosis rate, whereas the kidneys have a high rate of phagocytosis but it takes 125 h to reach the half maximum.

We performed a sensitivity analysis to determine what parameters are most influential on the model predictions. First, we measured the

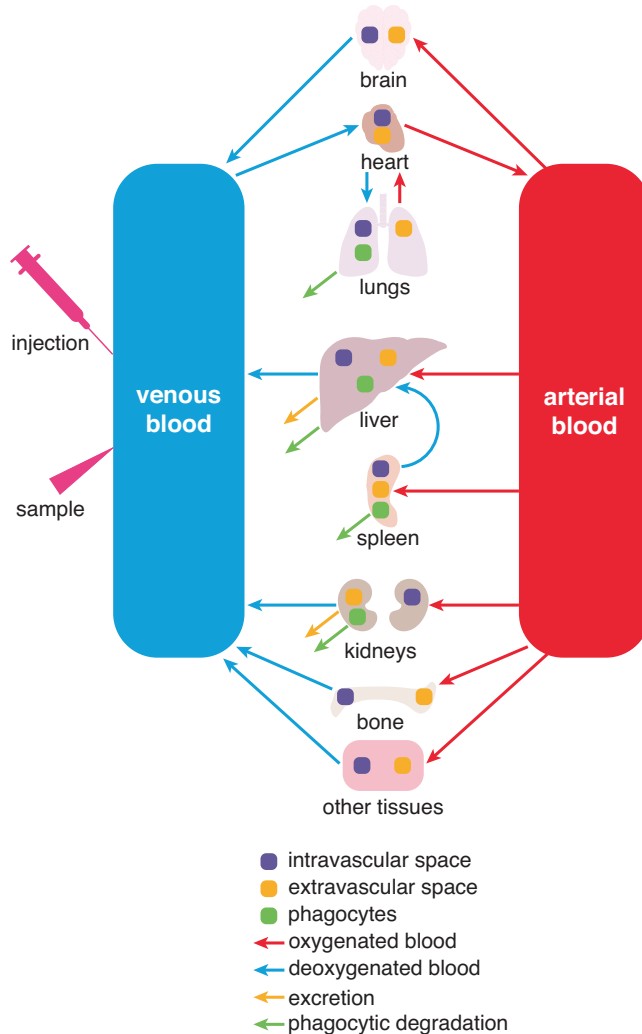

**Fig. 3 | Schematic of PBPK model describing PACE NP biodistribution in blood and various organs.** The other tissues compartment is defined as all tissues separate from the ones displayed here.

change in blood concentration generated by a perturbation of each parameter. We performed single simulations of an NP dose of 0.5 mg and computed the area-under-the-curve (AUC) of the blood concentration. Each parameter was perturbed by 1%, and the resulting change in AUC was measured and divided by the change in the parameter (denoted P), to calculate the normalized sensitivity coefficient (NSC)[18–20]:

$$NSC = \frac{\left(\frac{AUC - AUC_0}{AUC_0}\right)}{\left(\frac{P - P_0}{P_0}\right)}, \qquad (1)$$

For which the subscript 0 denotes the baseline parameter value ($P_0$) and AUC ($AUC_0$). The NSC values for each parameter are included in Supplementary Table 3.

For all but three parameters, the NSC was equal to 0, indicating that the NP pharmacokinetics (blood concentration) are practically insensitive to changes in these parameters. Thus, these parameters only appear to significantly impact the biodistribution to individual organs, not the NP concentration in the blood. In the case of $(K_{max})_{Liver}$, the NSC < 0, which implies that the blood concentration is somewhat sensitive to changes in $(K_{max})_{Liver}$ and that increasing this parameter would consequently reduce the blood concentration AUC. However, |

|NSC| <0.5, which implies that the model is not acutely sensitive to changes in this parameter. This is not unexpected, as the model assumes a distribution of parameter values will be used to construct a probabilistic model output. By far, the highest NSC value corresponds to $K_{bile}$, which is assumed to control the rate of PACE-PEG NP excretion because $K_{urine} = 0$. For $K_{bile}$, |NSC| = 0.6, meaning that small fluctuations in the rate of bile excretion greatly impact the blood NP concentration, as compared to other parameters.

Some of the parameter distributions (defined by the mean and standard deviation for each parameter in Supplementary Table 3) are wider than others, such that different parameters have different constraints in terms of how much they can change and still generate a model output that fits the experimental data. As a second measure of sensitivity, we calculated the coefficient of variation (CV) for each parameter, wherein we divided the standard deviation of the parameter value by its corresponding mean, to estimate how sensitive the model-data fit is to changes in each of these parameters (column in Supplementary Table 3). The CV analysis corroborates results from the NSC analysis; namely, $K_{bile}$ can vary the least (low CV) and still produce a result that fits the experimental data. More robust than the NSC analysis, CVs for $(K_{max})_{Liver}$, $PA_{Bone}$ and $PA_{Heart}$ vary the least and reproduce experimental results. The former two parameters are the sole controllers of NP uptake by the bone and heart, respectively, thus a low CV is consistent for these variables.

### Decoy occupation of phagocytic cells

Encouraged by our finding that increasing the dose of PACE NPs to limit clearance by the RES affects both circulation half-life and enhances biodistribution beyond the liver and spleen—as well as our modeling results confirming the importance of macrophage phagocytosis in controlling biodistribution—we next sought to determine whether pre-administering agents to intentionally deplete or occupy macrophages in the liver and spleen could have a similar effect (Fig. 5a). Others have described decoy strategies with various NP systems[7–9]; these strategies include pre-administering lipid emulsions or unloaded polymeric NPs, as well as saturating RES cells with red blood cells (RBCs) by administering an anti-erythrocyte antibody. Prior reports have described the use of these tools individually, but they have never been compared for their effectiveness. Here, we directly compared alternate decoy strategies to evaluate their ability to modulate blood concentration and biodistribution of the same NP population. As ~230 nm PACE NPs accumulate primarily in the liver and spleen without intervention, we tested whether prior IV administration of intralipid (20% emulsion), blank NPs composed of poly(lactic-co-glycolic acid) (PLGA), or anti-RBC antibodies influenced PACE NP blood concentration and biodistribution. We compared these effects to NPs administered to mice who received clodronate liposomes, as a positive control, because these mice have temporarily depleted macrophage populations (Supplementary Fig. 4). Anti-RBC antibodies were administered 12 h before PACE NP administration whereas intralipid, blank PLGA NPs, and clodronate liposomes were administered 24 h prior to PACE NP administration (Fig. 5b). In each case, we found that pre-administration of decoys modulated both blood concentration and biodistribution compared to PACE NPs administered alone in the absence of a decoy. The effect on blood concentration and resulting half-life is most apparent at early timepoints, with NPs being cleared from the blood by 48 h in all cases (Fig. 5c). Clodronate liposomes had the most pronounced effect on blood concentration and biodistribution (Fig. 5c–k), resulting in the highest NP accumulation in the lungs. These animals also exhibited increased NP uptake in the heart, kidneys, and bone marrow compared to the PACE NP only control group. Surprisingly, both intralipid and blank PLGA NP pretreatment decreased circulation half-life compared to PACE NPs alone (Fig. 5c). However, both of these decoys modulated biodistribution

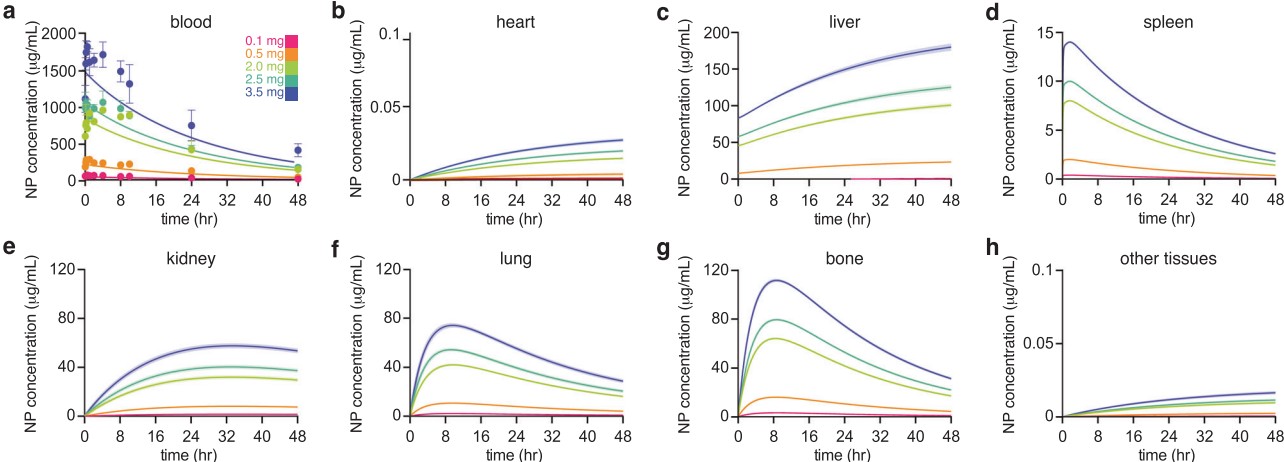

**Fig. 4 | PBPK Modeling of PACE-PEG NP Biodistribution. a** Model prediction of NP blood concentration for various NP doses (0.1 mg: pink, 0.5 mg: orange, 2.0 mg: green, 2.5 mg: turquoise, and 3.5 mg: indigo per animal corresponding to approximately 30 billion, 160 billion, 650 billion, 800 billion, and 1 trillion NPs, respectively) over time with experimental datapoints overlaid (*n* = 3 mice per group per time point; error bars represent standard deviation (SD)). Solid lines indicate the mean of the model output, shaded area indicates the +/− standard error of the mean (SEM). Colors correspond to dosages simulated in the model, compared to corresponding data. **b** Model prediction of NP tissue concentration over time in the heart. **c** Model prediction of NP tissue concentration over time in the liver. Solid lines indicate the mean of the model output, shaded area indicates +/− SEM.

**d** Model prediction of NP tissue concentration over time in the spleen. Solid lines indicate the mean of the model output, shaded area indicates +/− SEM. **e** Model prediction of NP tissue concentration over time in the kidneys. Solid lines indicate the mean of the model output, shaded area indicates +/− SEM. **f** Model prediction of NP tissue concentration over time in the lungs. Solid lines indicate the mean of the model output, shaded area indicates +/− SEM. **g** Model prediction of NP tissue concentration over time in the bone marrow. Solid lines indicate the mean of the model output, shaded area indicates +/− SEM. **h** Model prediction of NP tissue concentration over time in the brain and other tissues (combined). Solid lines indicate the mean of the model output, shaded area indicates +/− SEM.

(Fig. 5d–f, i), increasing uptake in the heart, lung, and spleen. Clodronate liposomes, intralipid, and blank PLGA NPs all exhibited increased fluorescent signal in the liver compared to PACE NPs alone (Fig. 6). Flow cytometry analysis of cell types in the liver revealed that decoy treatments result in near-full occupation of F4/80⁺ macrophages and elevated uptake in other cell types, such as CD31⁺ endothelial cells (Fig. 5g, j). These results support the hypothesis that Kupffer cell occupation in the liver enables uptake in additional cell and tissue types. The decoy with the most unique biodistribution signature was anti-RBC antibodies. While this cohort exhibited broader biodistribution, similarly to other decoys, primary PACE NP accumulation occurred in the spleen, with significantly reduced accumulation observed in the liver (Fig. 5e). The spleens of these animals were enlarged, suggesting splenomegaly due to extravascular hemolysis. In splenomegaly, there is increased pooling of RBCs in the spleen[21]. This, combined with increased phagocytic activity of spleen macrophages due to antibody-bound RBCs, may explain the increased NP accumulation in the spleen. The anti-RBC pretreated cohort also exhibited the highest PACE NP accumulation in the bone marrow. This may be due to increased blood flow to the bone marrow as it increases production of immature RBCs in response to hemolytic anemia. It has been shown that anemia in rats leads to tripling of blood flow to femoral bone marrow[22].

### Therapeutic potential of decoys for nucleic acid delivery

From these observations, we hypothesized that decoy administration could be used to enhance therapeutic delivery for diseases by improving NP accumulation in tissues and cells of interest. In particular, knockdown of Pcsk9 in liver hepatocytes is a well-known strategy for the treatment of hypercholesterolemia[23,24] and relies on successful delivery of knockdown agents to hepatocytes. Using pre-administration of PLGA NP decoys, we observed enhanced PACE NP accumulation in hepatocytes compared to PACE NP administered alone (Figs. 6d, 7). We formulated PACE NPs encapsulating siRNA against Pcsk9 and administered these into C57BL/6 mice via retro orbital injection with or without pre-administration of blank PLGA NP

decoys. We quantified the Pcsk9 level in hepatocytes and measured blood cholesterol levels. PLGA NP decoy pre-administration significantly increased the level of Pcsk9 knockdown (85 ± 8%) compared to PACE NPs administered in the absence of decoy (25 ± 4%) (Fig. 7). Measurements of blood cholesterol revealed reduced cholesterol levels in the decoy group (76 ± 6 mg/dL) compared to the PACE NPs only group (89 ± 4 mg/dL) and the control group (100 ± 8 mg/dL). These results suggest that pre-administration of PLGA NP decoys enables siRNA-loaded PACE NPs to avoid uptake by Kupffer cells and instead accumulate in target hepatocytes resulting in a more robust therapeutic effect.

### Improvements in NP design: antibody-mediated targeting

Beyond broadly improving NP accumulation in non-RES cells, enhancing uptake in specific cell types remains of interest for many diseases. Efforts to conjugate targeting moieties to the surface of NPs have been described extensively in the literature but may still require enhanced circulation time to allow time for interaction with their intended target. We hypothesized that increasing the circulation time of Ab-targeted PACE NPs would improve their targeting capability. To test this, we synthesized a PACE-PEG-MAL polymer with a maleimide group capable of thiol chemistry for conjugation to a monobody adapter, and subsequently a targeting antibody, as has previously been described (Fig. 8a)[25]. We formulated dye-loaded NPs composed of 50:50 PACE-PEG:PACE-PEG-MAL and attached anti-human-CD4 antibodies to the surface (CD4 NPs). We then incubated these NPs with whole human blood for 5 min, 12 min, 30 min, or 1 h, and compared their binding to that of NPs with an attached isotype-matched Ab as a control (isotype NPs). We observed immediate and sustained targeting of isolated CD4⁺ cells, with greater than 90% of CD4⁺ cells taking up CD4 NPs across all time points (Fig. 8b). Rapid targeting of CD4⁺ cells suggested this NP formulation may be successful in vivo with reasonable extensions to blood circulation time. Both control NPs and CD4 NPs were equally taken up by leukocytes as a whole, and this uptake increased over time (Fig. 8c). These results suggest that both control NPs and CD4-NPs are capable of passive uptake in bulk leukocytes, but only CD4-targeted

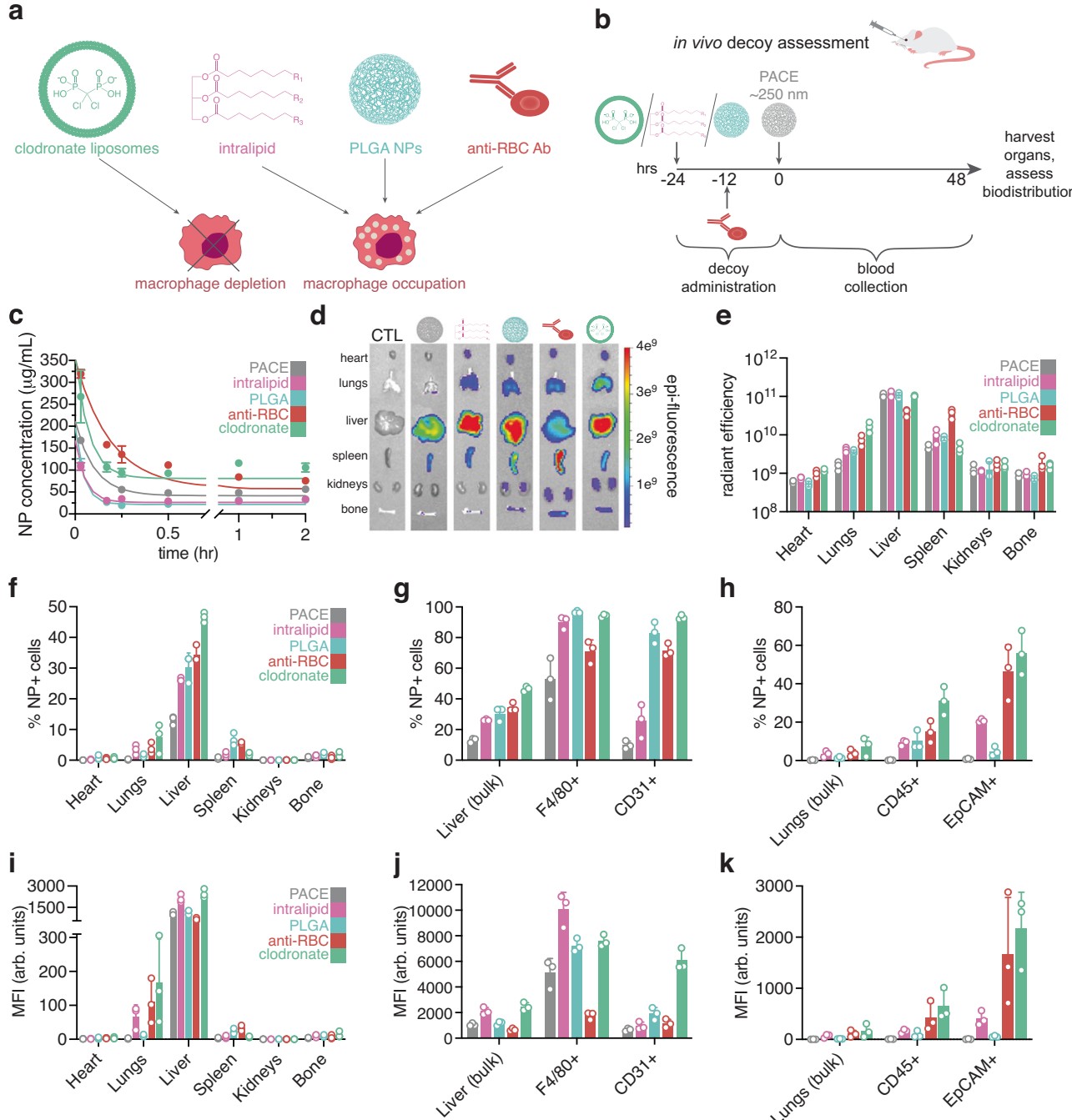

**Fig. 5 | Effects of decoy pre-administration on PACE NP blood concentration and biodistribution. a** Schematic of decoys to deplete or occupy macrophages in the liver and spleen. **b** Schematic of decoy pre-administration scheme. **c** Blood NP concentration over time following intravenous injection of dye-loaded PACE NPs alone (gray) or with decoy pre-administration (intralipid: pink, PLGA NPs: cyan, anti-RBC Ab: red, or clodronate liposomes: mint green) using quantitative microscopy ($n$ = 3 mice per group per time point; error bars represent standard error of the mean (SEM)). Representative end-point (**d**) IVIS analysis of PACE NP uptake in organs (heart, lungs, liver, spleen, kidneys, and bone) alone or following decoy-pre-administration, CTL: untreated control. End-point analysis of (**e**) whole organ fluorescence quantification of PACE NP uptake in organs ($n$ = 3 mice per group per organ; error bars represent SEM) alone or following decoy pre-administration. End-point analyses of (**f**) %NP+ cells in homogenized organs by flow cytometry ($n$ = 3 mice per group per organ; error bars represent SEM), (**g**) %NP+ cells in homogenized liver populations (bulk, F4/80$^+$, and CD31$^+$) by flow cytometry ($n$ = 3 mice per group per population; error bars represent SEM), and (**h**) %NP+ cells in homogenized lung populations (bulk, CD45$^+$, and EpCAM$^+$) by flow cytometry ($n$ = 3 mice per group per population; error bars represent SEM). End-point analyses of (**i**) Mean fluorescence intensity (MFI) in arbitrary units (arb. units) of homogenized organs by flow cytometry ($n$ = 3 mice per group per organ; error bars represent SEM), (**j**) MFI in homogenized liver populations (bulk, F4/80$^+$, and CD31$^+$) by flow cytometry ($n$ = 3 mice per group per population; error bars represent SEM), and (**k**) MFI cells in homogenized lung populations (bulk, CD45$^+$, and EpCAM$^+$) by flow cytometry ($n$ = 3 mice per group per population; error bars represent SEM).

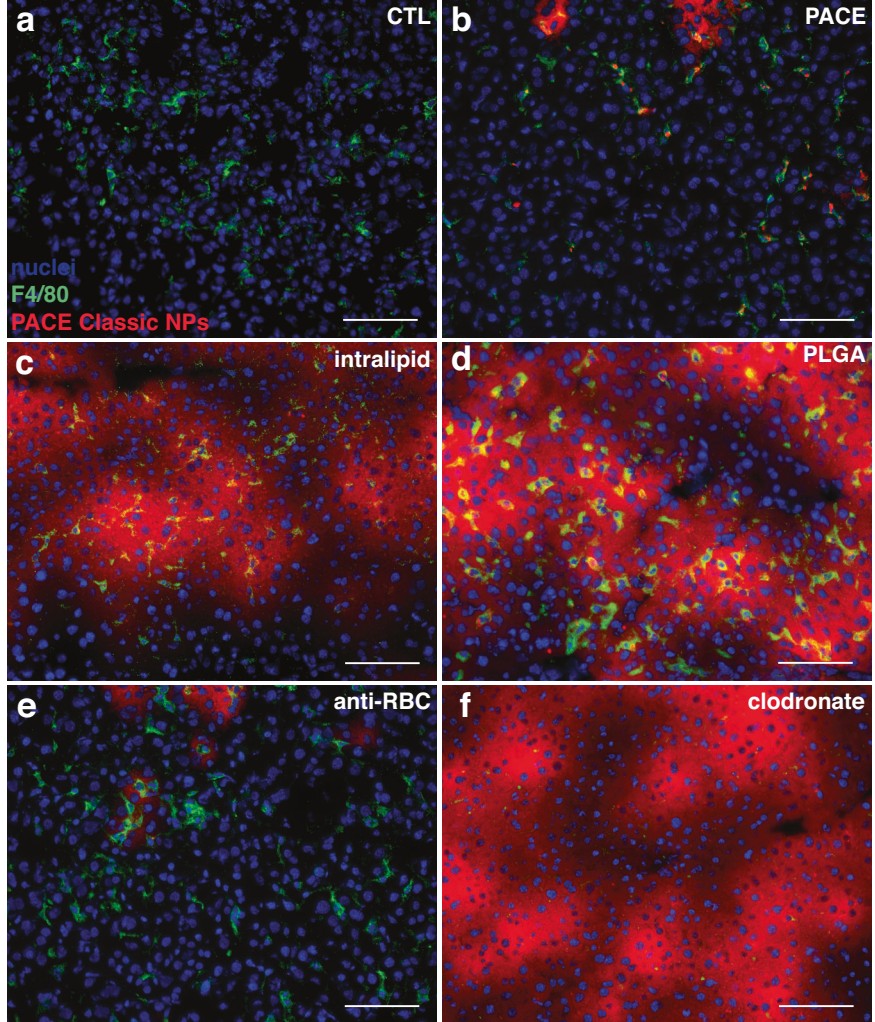

**Fig. 6 | Effects of decoy pre-administration on PACE NP distribution in the liver.** Representative epifluorescence liver images from three independent experiments of (**a**) untreated control animals, (**b**) PACE NP-treated animals, (**c**) intralipid and PACE NP-treated animals, (**d**) PLGA NP and PACE NP-treated animals, (**e**) anti-RBC Ab and PACE NP-treated animals, and (**f**) clodronate liposome and PACE NP-treated animals. Nuclei: blue, F4/80$^+$ macrophages: green, PACE NPs: red. Scale bars, 100 μm.

NPs accumulate significantly in CD4$^+$ T-cells in peripheral blood and do so in a realistic time frame.

We next studied the in vivo targeting potential of PACE-PEG-MAL-based NPs in rats with decoy pre-administration. Either isotype control or rat CD4-targeting NPs were administered to wildtype rats 24 h after clodronate liposome administration (Fig. 8d). Peripheral blood mononuclear cells (PBMCs) were isolated from peripheral blood samples 2 h later and analyzed for NP uptake by flow cytometry. Similar to the ex vivo results observed in human blood, CD4-targeted NPs accumulated preferentially in CD4$^+$ T-cells (Fig. 8e,f). Together, these results demonstrate the in vivo targeting capability of PACE-PEG-MAL NPs aided by decoy pre-administration.

## Discussion

Optimization of in vivo delivery by non-viral nanocarriers is complex, as these carriers face biological barriers at multiple length scales. Disease state is also important to consider, as pathophysiology can impact NP trafficking. We have shown that both polymer chemistry and NP size influence time-based circulating blood concentration and biodistribution, though the relationship between these two metrics remains unclear. Does increased time in circulation lead to more dispersed biodistribution? From our results, this may be true when chemistry or size has a major impact on circulation half-life (as is the

case with PEGylated PACE NPs as compared to PACE NPs). Interestingly, even with the significant increases in circulation half-life and more widespread distribution—obtained by either dosing or decoy administration—accumulation in the liver and spleen is barely reduced. We believe the phagocytic activity in these organs is still active even if at a slower rate or reduced capacity, allowing additional opportunities for contact with other cell and tissue types. For example, we observe increased accumulation in other tissues such as the lungs, kidneys, and bone marrow, as well as other cell types apart from macrophages in both the liver and the lungs. These results also highlight the importance of examining multiple aspects of in vivo behavior (blood concentration, whole organ distribution, and uptake in individual cells) simultaneously to gain the most comprehensive perspective of NP biodistribution.

The data discussed above are strengthened by a pharmacokinetic model of the underlying processes, which can predict NP concentration in various tissues at different time points after administration. Through multiple sensitivity analyses, the PBPK model developed here suggests that the rate of NP excretion in bile is the most influential on pharmacokinetics and biodistribution. This is echoed by the study from which we developed the model source code[17]; 100 nm AuNPs showed a similar sensitivity ($|NSC = 0.6|$) to the rate of NP excretion in bile. Other PBPK modeling studies of different formulations of gold

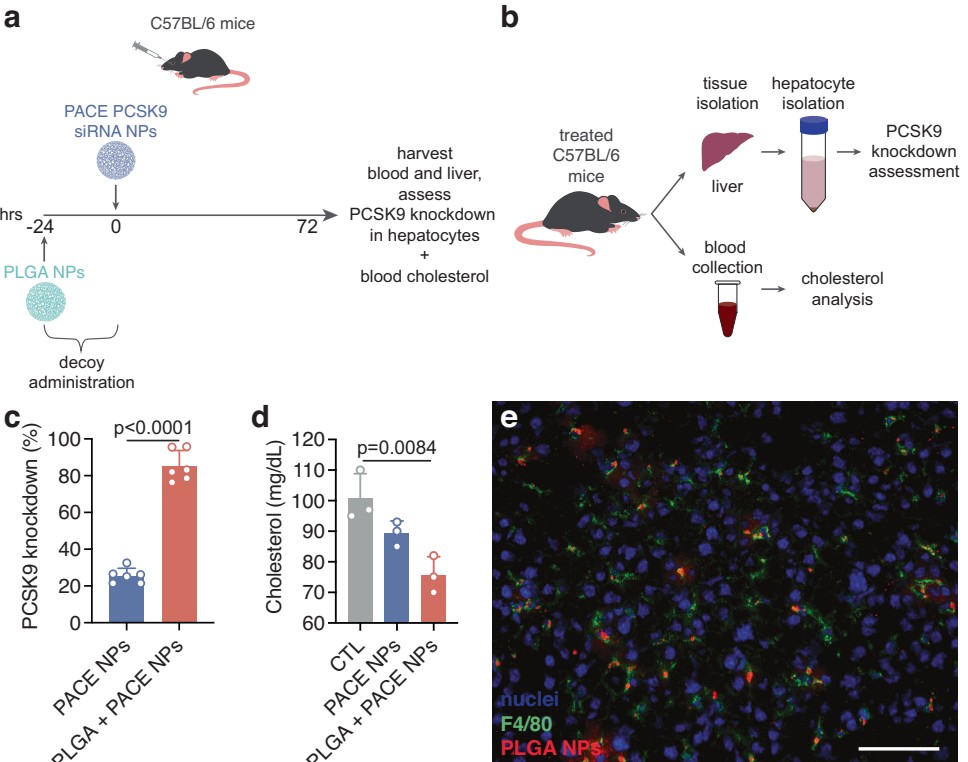

**Fig. 7 | PLGA decoy pre-administration significantly enhances PCSK9 knockdown by PACE siRNA NPs in hepatocytes. a** Schematic of PLGA NP decoy pre-administration scheme with PCSK9 siRNA-loaded PACE NPs and in vivo assessment workflow. **b** Schematic of liver and blood end-point analyses to assess PCSK9 activity. **c** %PCSK9 mRNA knockdown as measured by quantitative RT-PCR in hepatocytes of PACE PCSK9 siRNA NP-treated mice and PACE PCSK9 siRNA NP-treated mice following PLGA NP pre-administration ($n = 6$ mice per group per population; error bars represent standard error of the mean (SEM)). An unpaired two-tailed $t$ test was used for statistical analysis; $p = 0.000000024$, $t = 15.601510$,

df = 10. **d** Cholesterol levels in untreated control, PACE PCSK9 siRNA NP-treated mice, and PACE PCSK9 siRNA NP-treated mice following PLGA NP pre-administration ($n = 3$ mice per group per population; error bars represent SEM). An ordinary one-way ANOVA was used for statistical analysis ($F = 11.85$, $R^2 = 0.7980$, $p = 0.0082$) with Holm-Šídák's post-test for multiple comparisons; for untreated control (CTL) vs. PLGA + PACE NPs, $p = 0.0084$. **e** Representative epifluorescence liver image from three independent experiments of dye-loaded PLGA NP-treated animals 24 h post-IV administration. Nuclei are shown in blue, F4/80+ macrophages are shown in green, and PLGA NPs are shown in red. Scale bar, 100 μm.

NPs in rats contrasted these results, indicating that the NP bile excretion rate mainly impacts the GI compartment, not the blood[18]. However, these results were developed based on data from gold NPs with diameters of 5 nm. Since the NPs used in the current study as well as the NPs in the study used to parameterize the model have diameters ≥100 nm, it is possible that the size difference accounts for this difference in sensitivities.

With these model results come model limitations; importantly, the results generated by the model presented here are uncertain in that the biodistribution of NPs throughout the organ compartments is calculated based on flow cytometry, which uses NP fluorescence as a means of estimating relative amounts of NPs in different organs or cell types. As a result, the biodistribution data used to parameterize our model is based on the ratio of non-liver organ fluorescence and liver fluorescence. This naturally predisposes the model to error, as it is unclear the actual amount of NPs accumulating in the liver, and even small errors in this calculation predispose the model to large errors in the other organ compartments, both in their absolute value and kinetics. Nevertheless, the model fits the experimental data and provides insight into the system. In Fig. 4, the model blood concentration profile underestimates the experimental values of NP concentration in the blood as the dose is increased, suggesting that the model overestimates the amounts taken up in the organs. As a result, the model fails to reproduce the relationship between PACE-PEG NP half-life and dose seen in Fig. 2. This result suggests that a *non-modeled* effect (i.e., phagocytic capacity) causes the model to underestimate the blood concentration. This model error demonstrates the utility of a

simplified model in describing the complex process of NP pharmacokinetics and biodistribution, by indicating that a limiting capacity of phagocytes may explain the discrepancy between the model and experimental results. That said, the model used in this study mainly provides a means of hypothesis generation and must be further improved and tested in subsequent studies. Importantly, future iterations of the model will investigate the mechanism of accumulation of NPs in the blood within the first 8 h post-administration, which is the primary source of error in the predicted blood pharmacokinetic profile.

By using dose to increase circulation time instead of making changes to the NP, we can more directly link increased circulation time to more widespread NP accumulation. We found that increasing the dose of PACE-PEG NPs significantly increases circulation half-life up to ~4 fold. This increased circulation time led to increased biodistribution and uptake within individual cells in multiple tissues beyond the liver and spleen once the dose increased beyond a threshold. This threshold, which may be around 2.5 mg/mL of polymeric NPs, is indicated by the continued increase in liver macrophages until this dose is reached, signifying they are at capacity. These results suggest that polymer/NP characteristics are not the only parameters that one can manipulate to achieve desired tissue tropism; NP dosage is also influential. More importantly, they also suggest that macrophages in the liver and spleen appear to have a limited, time-based capacity for phagocytosis of polymeric NPs, which, if exceeded, can enable access to additional tissues and cell types. This finding supports the strategy to intentionally occupy macrophages with something other than a therapeutic NP

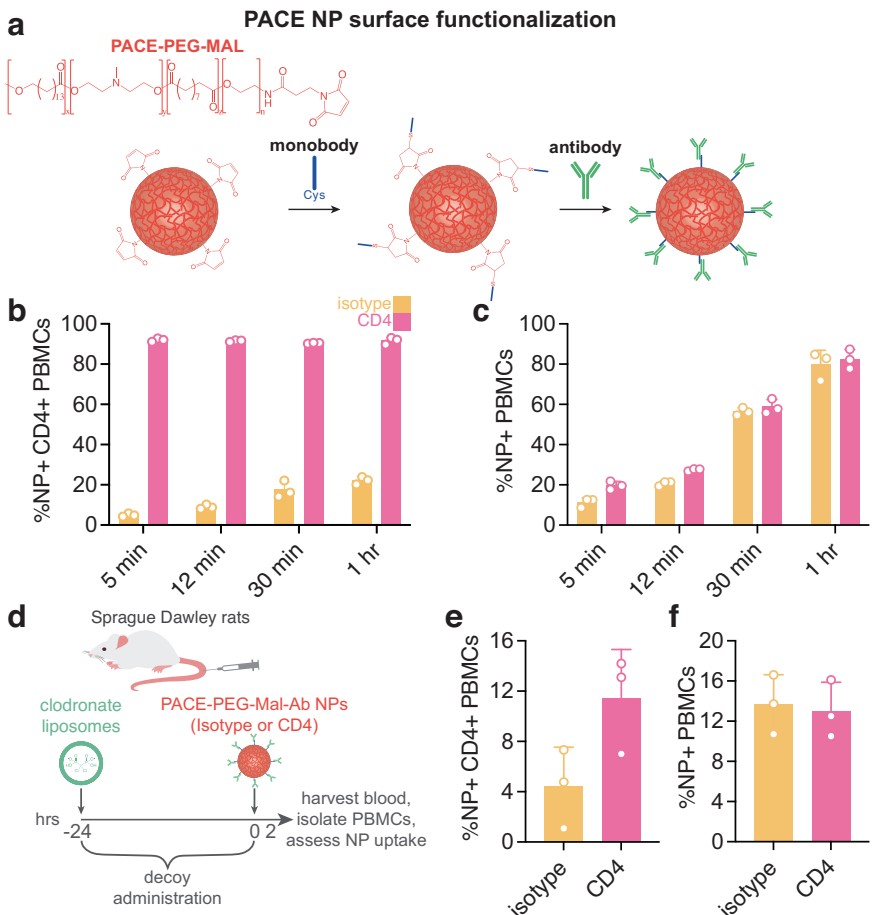

**Fig. 8 | CD4-targeting PACE NPs accumulate selectively in CD4⁺ T-cells in vitro in human blood and in vivo in rats. a** Schematic of PACE-PEG-maleimide NP monobody and antibody conjugation process. **b** %NP + CD4⁺ PBMCs from human blood following in vitro incubation with either isotype control (yellow) or CD4-targeting (pink) DiD-loaded PACE-PEG-maleimide NPs for 5 min, 12 min, 30 min, and 1 hr ($n = 3$ blood samples per group per time point; error bars represent standard error of the mean (SEM)). **c** %NP+ PBMCs from human blood following in vitro incubation with either isotype control or CD4-targeting DiD-loaded PACE-PEG-maleimide NPs for 5 min, 12 min, 30 min, and 1 hr ($n = 3$ blood samples per group per time point; error bars represent SEM). **d** Schematic of in vivo CD4⁺ T-cell PACE-PEG-maleimide NP targeting experiment with clodronate liposome decoy pre-treatment in rats. **e** %NP + CD4⁺ PBMCs from rat blood two hours after intravenous delivery in vivo of either isotype control or CD4-targeting DiD-loaded PACE-PEG-maleimide NPs ($n = 3$ rats per group; error bars represent SEM). **f** %NP+ PBMCs from rat blood two hours after intravenous delivery in vivo of either isotype control or CD4-targeting DiD-loaded PACE-PEG-maleimide NPs ($n = 3$ rats per group; error bars represent SEM).

(which may be limited in possible dose) could offer opportunities to manipulate in vivo delivery of the therapeutic payload.

Indeed, we found that pre-administration of decoy formulations to eliminate or occupy macrophages primarily in the liver can be used to modulate both blood concentration and biodistribution of a second NP formulation. Interestingly, not all decoy formulations increased the half-life of PACE NPs, but each did result in altered and more widespread distribution, further supporting that it is not one parameter (half-life) that controls NP accumulation outside the phagocytes of the liver and spleen.

Using parameters and decoy strategies that result in specific tissue tropism, we were able to design two therapeutic delivery strategies. One strategy in particular, the administration of PLGA decoy NPs, may have important therapeutic applications. PLGA NPs, which are themselves well-tolerated, provide a means to shift therapeutic NPs from macrophages to other cell types in the liver with minimal long-term interference. We demonstrate that delivery of siRNA against Pcsk9 using PACE NPs, administered after a decoy dose of PLGA, resulted in significantly better knockdown and reduced blood cholesterol levels than when administered in NPs without decoy pre-administration.

Lastly, we coupled decoy pre-administration with antibody-conjugated PACE NPs to achieve NP uptake in a specific cell

population in vivo. This combined strategy was successful in increasing NP accumulation in peripheral blood CD4⁺ T-cells, demonstrating the ability to employ multiple approaches to manipulate NP biodistribution for a target cell type beyond base polymer chemistry and NP characteristics. Importantly, these strategies should be broadly applicable to other non-viral vehicles beyond the polymeric NPs studied here.

We believe this is only the beginning of identifying key parameters to optimize systemic in vivo delivery. Further exploration of the mechanisms of decoy-mediated macrophage occupation will be required for more precise manipulation, with particular interest in NP administration timing and dose. Moreover, improvements and additions to decoy formulation and therapeutic NP design may be able to capitalize on extended circulation time to intentionally accumulate in target sites, rather than simply delaying accumulation by phagocytes. Regarding the potential for clinical translation, the safety of decoy formulations in addition to therapeutic PACE formulations will need to be assessed. Temporary phagocytic occupation with biocompatible materials as opposed to depletion of phagocytes is the most likely strategy to succeed in the clinic. Both intralipid and PLGA NPs already have favorable safety profiles, as intralipid is already used in the clinic and PLGA particles are part of several FDA-approved products for drug delivery.

## Methods

### Materials

Bovine serum albumin (BSA) and ultra-pure dimethyl sulfoxide (DMSO) were purchased from AmericanBio. DiD dye was purchased from Biotium and dissolved in DMSO at 10 mg/mL prior to use. Monoclonal antibodies for flow cytometry were purchased from ThermoFisher Scientific (FITC anti-mouse CD45 (30-F11), PE anti-mouse CD31 (390), anti-human CD4 (9H5A8), anti-rat CD4 (W3/25), PE anti-rat CD3 (G4.18)) and Biolegend (PE anti-mouse F4/80 (BM8), PE anti-mouse CD326 (G8.8), PE anti-human CD3 (UCHT1), FITC anti-human CD8a (RPA-T8), FITC anti-rat CD8a (OX-8)). Anti-mouse RBC monoclonal antibody (34-3 C) was purchased from Hycult Biotech. Clodronate liposomes were purchased from Liposoma. Collagenase Type I was purchased from Worthington Biochemical Corporation. Heparinized Eppendorf tubes, 1X RBC Lysis Buffer, 1X Hank's Balanced Salt Solution (HBSS), and Fisherbrand Superfrost Microscope slides were purchased from ThermoFisher Scientific. Heparin (1000 USP/mL) was purchased from Cardinal Health. ON-TARGETplus Mouse Pcsk9 siRNA was purchased from Horizon Discovery Biosciences Ltd (Target Sequence UGUCUAUGCCGUCGCGAGA). Glass-bottom 384-well plates were purchased from MatTek Life Sciences. Hoescht 33342 was purchased from Tocris Bioscience. Ethylenediaminetetraacetic acid disodium salt dihydrate (EDTA), DNase I (Protein ≥ 85%, ≥ 400 Kunitz units/mg protein), Intralipid (20% emulsion), isoflurane, 15-Pentadecalactone (PDL, 98%), diethyl sebacate (DES, 98%), sebacic acid (SA, 98%), N-methyldiethanolamine (MDEA, 99 + %), polyethylene glycol (PEG, 5000 MW), diphenyl ether (99%), immobilized Candida antarctica lipase B (CALB) supported on Novozym 435, chloroform (HPLC grade), dichloromethane (HPLC grade, 99 + %), and hexanes (HPLC grade, 97 + %) were all purchased from Millipore Sigma and used as received.

### Poly(amine-co-ester) synthesis and purification

PACE polymers were synthesized and characterized as described previously[10]. Briefly, PDL was copolymerized with MDEA and DES (PACE) or SA (PACE-COOH) via a CALB enzyme catalyst in diphenyl ether with a feed molar ratio of 60:20:20 for PDL:DES/SA:MDEA. The reaction proceeded in two phases: an oligomerization phase at 90 °C under 1 atm of argon gas for 18–24 h, and a polymerization phase at 90 °C under vacuum for 48–72 h. For the synthesis of PACE-PEG, 5000 molecular weight (MW) PEG was added as an additional monomer at 34 weight % to PACE synthesis. The synthesis was otherwise identical. For PACE-PEG-maleimide synthesis, 5 K MW PEG-maleimide was added as an additional monomer similar to PACE-PEG synthesis[10]. After synthesis, diphenyl ether solvent was removed via hexane washes, and PACE polymer was dissolved in dichloromethane or chloroform and filtered. Solvents were removed using a rotary evaporator, and polymers were stored at −20 °C.

### NP formulation and characterization

PACE NPs loaded with DiD or unloaded (blank) PLGA NPs were formulated using an emulsion solvent evaporation technique as described previously[26]. Briefly, 50 mg of the polymer was dissolved in 1 mL of chloroform, DCM, or a 1:1 mixture of DCM and Ethyl Acetate overnight. 25 μL of DiD dye at 10 mg/mL in DMSO was added to the polymer solution before formulation at 0.5% (w/w). The resulting solution was emulsified under vortex with 2 mL of 5% (w/v) low MW polyvinyl alcohol (PVA) in diH$_2$O or diH$_2$O alone and sonicated with a probe tip sonicator (Sonics) and then diluted into 10 mL of 0.3% (w/v) PVA in diH$_2$O or diH$_2$O alone while stirring. Any remaining organic solvent was evaporated using a rotary evaporator. NPs formulated with PVA were centrifuged (30 min, 18,500 × g, 4 °C) and resuspended in fresh diH$_2$O after removing the supernatant twice to remove excess PVA. NPs formulated in diH$_2$O without PVA were concentrated by centrifugation (30 min, 21,000 × g, 4 °C). PACE NPs encapsulating siRNA against Pcsk9 were similarly formulated, but in this case Pcsk9 siRNA was first dissolved in sodium acetate buffer (25 mM, pH 5.8) and emulsified with the polymer solution and sonicated using a probe tip sonicator. This emulsion was then emulsified again with 2 mL of 5% (w/v) low MW PVA solution and sonicated. As above, this emulsion was then diluted into 0.3% (w/v) PVA while mixing, the remaining organic solvent was evaporated, and the NPs were washed with diH$_2$O by centrifugation (30 min, 18,500 × g, 4 °C). The size and zeta potential of all NPs were measured using dynamic light scattering (DLS, Malvern Instruments). NPs were then aliquoted and flash-frozen in liquid N$_2$ before storage at −80 °C before use. One of the aliquots was lyophilized and weighed to determine the NP concentration for each batch. A summary of NP formulation specifications and resulting characteristics can be found in Supplementary Table 1.

### In vivo mouse NP administration and mouse blood collection

All animal procedures were performed in accordance with the guidelines and policies of the Yale Animal Resources Center (YARC) and approved by the Institutional Animal Care and Use Committee (IACUC) of Yale University (IACUC Protocol Number 2020-11228). Following federal guidelines, animals were housed in an environment with a 12 h light/12 h dark cycle, a temperature of 20–26 °C, and 30–70% humidity. Food and water were given ad libitum. Male and female BALB/c mice aged 4–6 weeks old were used. For studies using a cystic fibrosis (CF) disease model, male and female mice homozygous for the F508del mutation (fully backcrossed C57/BL6 background) were used. All mice were genotyped before use. Mice were anesthetized via vaporizer induction using isoflurane. Once respirations reduced to 1 breath/second, PACE NPs were administered into the bloodstream via a retro-orbital injection. We have previously demonstrated that biodistribution following retro-orbital injection is equivalent to biodistribution following tail vein injection[13]. Once animals were awake, 2 μL blood samples were collected at 2 min, 10 min, 15 min, 30 min, 1 h, 2 h, 4 h, 8 h, 10 h, 24 h, and 48 h following PACE NP injection with a p10 pipette from a tail nick made with a sterile blade. Sterile gauze was applied with pressure to the wound after blood collection to stop further blood flow. Blood volumes were promptly mixed with 2 μL heparin (1000 USP/mL) solution and 7 μL DPBS in heparinized tubes and flash frozen on dry ice and stored at −80 °C until analysis. After 48 h, animals were sacrificed. A cardiac puncture was used to draw blood following euthanasia.

### Biodistribution end-point analyses

Following euthanasia in DiD-loaded PACE NP studies, animals were heart-perfused with heparinized DPBS (100 USP/mL). Organs (heart, lungs, liver, spleen, kidneys, and bone) were harvested and imaged using an In Vivo Imaging System (IVIS) (Perkin Elmer). Half of each organ was then either homogenized for flow cytometry analysis or frozen in optimal cutting temperature (OCT) compound for analysis of tissue sections. For flow cytometry analysis, hearts, spleens, and kidneys were homogenized into a single-cell suspension through a 70 μm cell strainer with RPMI 1640 culture medium containing 10% FBS, washed with DPBS following centrifugation (5 min, 300 × g, 4 °C) and removal of the supernatant. Samples were centrifuged again and resuspended in DPBS containing 2% bovine serum albumin (BSA) and Hoescht 33342 (5 μg/mL). To extract cells from the bone marrow, both ends of one harvested femur bone were cut with scissors, and the femur was flushed with RPMI 1640 culture medium containing 10% FBS using a 25 G needle and 1 mL syringe. Bone marrow cells were then washed with DPBS following centrifugation (5 min, 300 × g, 4 °C) and removal of the supernatant. Samples were centrifuged again and resuspended in DPBS containing 2% bovine serum albumin (BSA) and Hoescht 33342 (5 μg/mL). Livers and lungs were processed further to determine PACE NP distribution in individual cell populations. Livers and lungs were first sliced manually and then digested using a solution of DNase (1 mg/mL) and Collagenase Type I (5 mg/mL) in HBSS at 37 °C

for 30 min on an orbital shaker. The resulting liver cell suspension was then filtered through a 70 μm cell strainer and rinsed twice with HBSS containing 4.8% BSA and 2 mM EDTA. Cells were collected by centrifugation at $330 \times g$ for 5 min at 4 °C. The cell pellet was then suspended in 2 mL of 1X RBC Lysis Buffer (Invitrogen) for 2 min, rinsed with 5 mL DPBS containing 2% BSA, and then collected again by centrifugation as described. Following collagenase and DNase digestion, the resulting lung cell suspension was further separated with shearing through an 18 G needle, filtered through a 70 μm cell strainer, and rinsed twice with DPBS containing 0.5% BSA. Cells were collected by centrifugation at $330 \times g$ for 5 min at 4 °C. The cell pellet was then suspended in 2 mL of 1X RBC Lysis Buffer for 2 min, rinsed with 5 mL of DPBS containing 2% BSA, and then collected again by centrifugation as described. Cell suspensions from the liver and lung were stained for cell-specific markers using fluorescently labeled antibodies. Diluted cell suspensions containing $1 \times 10^6$ cells were stained with 5 μL (1:20 dilution) of antibody for 1 h on ice. Unbound antibodies were rinsed off using 1 mL of DPBS containing 2% BSA, and cells were collected by centrifugation. Stained cells were then suspended in DPBS containing 2% BSA and 5 μg/mL Hoescht. Uptake of DiD PACE NPs in single cells was analyzed by flow cytometry (BD LSRII Green) and compared to tissues harvested from untreated control animals. A description of this gating strategy can be found in Supplementary Fig. 5. Tissue biopsies from mice treated with PACE DiD NPs were flash-frozen in OCT, sectioned, and imaged. Tissue images were acquired using an EVOS FL Auto 2 Cell Imaging System with standard RFP, GFP, and Cy5 filters with an Olympus super-apochromat $20 \times /0.75$ NA objective.

## High-throughput half-life measurements

Following blood collection as described above, samples were thawed on ice and deposited into a glass-bottom 384-well plate (10 μL per sample per well). A set of standards with known DiD-loaded NP concentrations were prepared using serial dilutions in blood ranging from 0 μg/mL to 250 μg/mL or up to 2500 μg/mL as appropriate depending on the NP doses used in the experiment and then deposited onto the plate. 10 μL per standard per well). Nine images of all standard and sample wells of the 384-well plate were acquired using an EVOS FL Auto 2 Cell Imaging System with a Cy5 filter cube using an Olympus super-apochromat $20 \times /0.75$ NA objective. Images were taken in the middle of each well and raw image files were saved for data analysis.

## Mathematical Model of NP Physiologically-Based Pharmacokinetics (PBPK)

A physiologically based, compartmental model of NP pharmacokinetics was adapted from a previous model used to predict the biodistribution of PEGylated gold NPs in mice[17]. This model was selected for use as a base model structure because of the data used to construct the model; NPs used in this study were of similar size to the NPs used in our study (approximate hydrodynamic diameter of 100 nm), the organism (mouse) and administration route (intravenous injection) were the same as in our study, and the biodistribution of these NPs was quantified over the same period (48 h). Using this model as the base model structure for our study, we reparametrized the model based on data obtained from our PACE-PEG NP biodistribution studies (Fig. 2).

The six organs in which NP accumulation was evaluated (liver, heart, spleen, kidney, bone marrow, and lung) were modeled as compartments that are exposed to NPs through the systemic circulation. A schematic of the model is shown in Fig. 3. In addition to these compartments, compartments for the brain and other tissues (denoting the remaining tissue space beyond the named compartments, such as that provided by the skin and muscle) are included. We have previously shown that PACE NPs do not accumulate significantly in the brain[11], thus the brain tissue is not included in our experimental results. Nevertheless, the model from which we derived the model for this study included a brain compartment with associated permeability to NPs, thus we have kept this compartment with associated NP absorption. Physiological values of blood flow are denoted Q. The mass of NPs, denoted M, is distributed throughout the volume of the organ, denoted V. Within each organ compartment, vascular, tissue-resident, and phagocytic (in the case of the lung, liver, spleen, and kidney) subcompartments are used to model PACE-PEG NP uptake. Values of these parameters (blood flows and volumes of each organ) are included in Supplementary Table 2. Differential equations describing the accumulation of NPs in the intravascular (subscript *vasc*), tissue-resident (subscript *extra*), and phagocytic (subscript *phago*) subcompartments, respectively, within each organ are as follows:

$$\frac{dM_{\text{vasc}}}{dt} = Q\left(\frac{M_{\text{Art}}}{V_{\text{Art}}} - \frac{M_{\text{vasc}}}{V_{\text{vasc}}}\right) - PA\left(\frac{M_{\text{vasc}}}{V_{\text{vasc}}} - \frac{1}{P}\frac{M_{\text{extra}}}{V_{\text{extra}}}\right) + R_{\text{rel}} - R_{\text{up}}, \quad (2)$$

$$\frac{dM_{\text{extra}}}{dt} = PA\left(\frac{M_{\text{vasc}}}{V_{\text{vasc}}} - \frac{1}{P}\frac{M_{\text{extra}}}{V_{\text{extra}}}\right) - R_{\text{elim}}, \quad (3)$$

$$\frac{dM_{\text{phago}}}{dt} = R_{\text{up}} - R_{\text{rel}}, \quad (4)$$

where PA is representative of the product of capillary NP permeability and regional blood flow, P is the tissue:plasma distribution coefficient, and the subscript Art indicates arterial blood. $R_{\text{up}}$, $R_{\text{rel}}$ and $R_{\text{elim}}$ denote NP uptake by phagocytic cells, NP release from the phagocytic cells, and elimination from the extravascular tissue in the form of bile (in the case of the liver) or urine (in the case of the kidney), respectively. These rates are calculated as:

$$R_{\text{up}} = \frac{K_{\text{max}} t^{n_H}}{K_{50}^{n_H} + t^{n_H}} M_{\text{vasc}}, \quad (5)$$

$$R_{\text{rel}} = K_{\text{rel}} M_{\text{phago}}, \quad (6)$$

$$R_{\text{elim}} = K_{\text{elim}} \frac{M_{\text{extra}}}{V_{\text{extra}}}, \quad (7)$$

With constants $K_{\text{rel}}$ and $K_{\text{elim}}$. In the case of the kidney, $K_{\text{elim}}$ is denoted $K_{\text{urine}}$, and in the case of the liver, $K_{\text{elim}}$ is denoted $K_{\text{bile}}$. All other organ compartments have a $K_{\text{elim}} = 0$. Equation 4 is a Hill equation describing the time-dependent mechanism of NP phagocytosis. $K_{\text{max}}$ denotes the maximum rate of phagocytosis, $K_{50}$ denotes the time required to reach half of the maximum uptake rate, and $n_H$ is the Hill coefficient. As in Eqs. 1–3, t denotes time. Equations 1–6 are representative of a single organ with a phagocytic sub-compartment and excretory capacity (i.e., the liver). The full system of differential equations is included in the supplementary material.

As stated previously, the model used in our study was originally developed using biodistribution data from PEGylated gold NPs in mice[17]. To fit the model to our PACE-PEG NP data, we re-parameterized the model by modifying the gold NP model's parameter values using MCIS[27]. MCIS involves running serial ($10^4$) simulations with the model, sampling parameter values from prescribed probability distributions, and updating the parameter distributions based on how well the simulation results fit the data. At the end of MCIS, parameters are represented as probability distributions, such that subsequent model simulations can be generated by sampling these parameter distributions to generate model results that are also probabilistic in nature. As such, the model not only provides a predicted value of NP concentration in the blood and tissue compartments, but also provides the probability that the model result lies within any specified interval. Beyond the utility of modeling PACE-PEG NP pharmacokinetics probabilistically, MCIS allows for the uncertainty of the experimental data

to be encapsulated within the model, which can be used to avoid erroneous predictions that are not properly backed by the experimental data.

We parameterized the model with MCIS using a three-step procedure. Firstly, we used MCIS to rescale the parameters used to model PEGylated gold NPs by probabilistically sampling from uniform distributions to determine the order of magnitude difference required to match the model to our PACE-PEG biodistribution data. Once we rescaled the parameters, we then performed another round of MCIS, sampling from tighter uniform distributions to finetune the parameter distributions. A final round of MCIS using normal distributions was used to construct the parameter values used in our study. With each step, the model result was compared with experimental data in order to determine how best to rescale and ultimately estimate the parameter distributions. The data used to parameterize the model included the PACE-PEG NP concentration in the blood at 0.03, 1, 8, 24, and 48 h, and the relative NP concentration in each non-liver organ at 3 and 48 h (excluding the 'other tissues' and brain compartments), normalized to the liver (endpoint biodistribution data). We discuss the full MCIS procedure in detail in the supplementary methods. The parameter values are tabulated in Supplementary Table 3.

The model was coded in R, as a system of ordinary differential equations. The R package deSolve was used to numerically solve the system of equations. The computations required for estimating the model parameters with MCIS, along with all model simulations, were performed on a personal laptop computer. All model code, including parameterization and simulations, is available at https://github.com/omrichfield/PACE-PBPK-Monte-Carlo_public.

To validate the model, we ran the model 100 times, sampling the parameters from the distributions defined in Supplementary Table 3. The model outputs were assessed by comparing the mean model output to the experimental data not used to parameterize the model (organ PACE-PEG biodistribution data at 6 and 24 h, and blood NP concentration data at 2 and 10 h). To appropriately compare the model results to the experimental data, we performed a Z test of the model-produced distribution (subscript mod) and the experimental data (subscript exp):

$$Z = \frac{\bar{x}_{mod} - \bar{x}_{exp}}{\sqrt{\frac{\sigma^2_{mod}}{n_{mod}} + \frac{\sigma^2_{exp}}{n_{exp}}}} \qquad (8)$$

Where $\bar{x}$ is the mean value, $\sigma$ is the standard deviation, and $n$ is the sample size. For model simulations, $n_{mod} = 100$. The Z test was performed for each of the 12 validation conditions described above. The Z score represents the number of standard deviations between the mean of the model prediction and the experimental mean. The typical acceptable range for PBPK model validation is a factor of two; namely, the model is considered validated if the model result is between 50% and 200% of the experimental mean[17,19]. Depending on the variance of the experimental data, this convention correlates approximately to |Z| < 20. As shown in Supplementary Fig. 3, the model error is mostly within this threshold. We also evaluated the model with an $R^2$ and root mean squared error (RMSE) for blood NP concentration at all time points at all five doses, and for organ biodistribution relative to the liver at all time points for a dose of 0.5 mg. These parameters and associated figures are included in Supplementary Fig. 3.

## siRNA delivery with decoy NPs

All animal procedures were performed in accordance with the guidelines and policies of the YARC and approved by the IACUC of Yale University (IACUC Protocol Number 2020-11228). Following federal guidelines, animals were housed in an environment with a 12 h light/12 h dark cycle, a temperature of 20–26 °C, and 30–70% humidity. Food and water were given ad libitum. PACE NPs encapsulating siRNA against Pcsk9 were administered to male and female C57BL/6 mice aged 4–6 weeks old via retro-orbital administration. Some animals received a decoy formulation pre-treatment, in which case 2 mg of blank PLGA NPs were administered retro-orbitally 24 h prior to PACE NP administration. All animals were euthanized 72 h following PACE NP administration. Blood was collected via cardiac puncture and sent to Antech Diagnostics for blood cholesterol analysis by standard protocols. Livers were collected to detect Pcsk9 mRNA levels. Following harvest, livers were further processed into single cell suspensions as described above. Hepatocytes were isolated from bulk liver cells by adapting a previously described protocol[28]. Briefly, a sample of liver cells from each animal was centrifuged at $50 \times g$ for 5 min at 4 °C. Total mRNA was isolated from hepatocyte samples using a Qiagen RNeasy Mini Kit. Isolated mRNA was used to synthesize cDNA using a High Capacity cDNA Reverse Transcription Kit (ThermoFisher). β-actin and Pcsk9 transcript levels were measured by quantitative real-time PCR using an Applied Biosystems StepOnePlus Real-Time PCR System instrument using the TaqMan Fast Universal PCR Master Mix (ThermoFisher). TaqMan primers for β-actin (Mm02619580_g1) and Pcsk9 (Mm01263610_m1) were used for gene expression analysis. All transcript levels were normalized to that of β-actin in the same sample.

## CD4 + T cell targeting ex vivo in human blood and in vivo in rats

Blood samples from consenting healthy donors were collected at the Yale School of Medicine. All collections were performed under protocols approved by the Yale University Institutional Review Board (IRB). Participants were not compensated for their enrollment. NPs were formulated following the procedure described above from a 50:50 blend of PACE-PEG polymer and PACE-PEG-maleimide polymer encapsulating DiD dye. These NPs were conjugated first to a monobody (Mb) adapter (clone FCM101)[29,30] via a thiol reaction between a thiol reactive group on the surface of the NPs (maleimide) and the single terminal cysteine of the Mb at a weight ratio of 10NPs:1 Mb. This reaction was carried out at room temperature for 1 h while shaking at 250 rpm. The excess Mb was removed by centrifugation (15 min, $18,000 \times g$, 17 °C). The NPs were redispersed in PBS at a concentration of 5 mg/mL and then coupled with antibodies against human CD4 (clone 9H5A8), rat CD4 (clone W3/25), or a nontargeting isotype control Ab at a weight ratio of 18NPs:1Ab at room temperature for 1 hr with gentle mixing. Excess Ab was removed with centrifugation (15 min, $18,000 \times g$, 17 °C). The resulting NPs were flash frozen and stored at −80 °C until use. This method is described in previously published reports[25]. Successful conjugation of the antibody was verified using a cell-based binding assay.

For human CD4 T cell targeting experiments, isotype control or CD4-targeted NPs were incubated with whole blood for 5 min, 12 min, 30 min, or 1 h. Nucleated cells were then isolated from the peripheral blood samples and processed for flow cytometry. Briefly, 200–300 µL of blood per sample was mixed with 50 µL of heparin prior to centrifugation for 3 min at $300 \times g$ and 4 °C. The supernatant was aspirated, and the remaining pellet was incubated with 1 mL of ACK lysis buffer and incubated for 3 min prior to centrifugation at $300 \times g$ and 4 °C (repeated twice). The pelleted cells were then resuspended in cold 1X DPBS and stained with antibodies for anti-human CD3 (clone UCHT1) and anti-human CD8a (clone RPA-T8). NP uptake in isolated and stained nucleated cells from peripheral blood was then analyzed by flow cytometry (BD LSRII).

For in vivo rat CD4+ T cell targeting experiments, female Sprague Dawley rats aged 3–5 weeks old were used. All animal procedures were performed in accordance with the guidelines and policies of the YARC and approved by the IACUC of Yale University (IACUC Protocol Number 2020-11228). Following federal guidelines, animals were housed in an environment with a 12 h light/12 h dark cycle, a temperature of 20–26 °C, and 30–70% humidity. Food and water were given ad libitum. Vasodilation in the tail was stimulated using a heat

lamp and clodronate liposomes (2 mL/kg) were administered via tail vein injection. CD4-targeting or isotype control PACE NPs (final concentration of 125 µg/mL in blood) were administered 24 h later following the same procedure via tail vein injection. 2 h after NP administration, rats were sacrificed. A cardiac puncture was used to draw blood following euthanasia. Nucleated cells were isolated from peripheral blood using the same procedure as described above for human blood. Samples were stained with antibodies for anti-rat CD3 (clone G4.18) and anti-rat CD8a (clone OX-8) prior to flow cytometry analysis using a BD LSRII cytometer.

### Data analysis
Results were analyzed using GraphPad Prism (version 10.2.2). For high-throughput quantitative microscopy analysis of NP blood concentration, raw images were analyzed using a custom MATLAB code as described previously[13]. Briefly, the intensity values of sample fluorophores (DiD encapsulated in PACE NPs) from each standard or sample image were summed and the background for each channel (blood only) was subtracted. Then, sample fluorophore intensity values for the images corresponding to each standard or sample were averaged. Standards were used to plot a linear standard curve in GraphPad Prism relating fluorescence intensity to PACE NP concentration and used to determine the concentration of PACE NPs in experimental samples. Concentration values for each blood sample per timepoint per animal were then plotted and fit to a one-phase decay curve in GraphPad Prism with the plateau constrained to zero to determine half-life values.

### Reporting summary
Further information on research design is available in the Nature Portfolio Reporting Summary linked to this article.

## Data availability
The data generated to support the findings in this study are contained in the article and the Supplementary Information. Raw images (blood quantification and IVIS) and FCS files can be provided upon request. All raw data for model development were generated in this study and can be found at: omrichfield/PACE-PBPK-Monte-Carlo_public: Public Physiologically Based Pharmacokinetic Model of polyamine coester nanoparticles (github.com, https://doi.org/10.5281/zenodo.10999309).

## Code availability
The model source code and data used to parameterize the model are available at: https://github.com/omrichfield/PACE-PBPK-Monte-Carlo_public: Public Physiologically Based Pharmacokinetic Model of polyamine coester nanoparticles (github.com, https://doi.org/10.5281/zenodo.10999309).

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

## Acknowledgements

We thank Shohei Koide and members of his laboratory at the New York University Langone Health Center for providing monobodies for this work. Descriptions of the monobody and its use in polymer NP work have been described previously[29,31,32]. This work was supported by grants from the National Institutes of Health (NIH; UG3/UH3 HL147352 to W.M.S. and R56 AI175206 to J.S.P. and W.M.S.) and the Cystic Fibrosis Foundation (CFF; EGAN641558 to M.E.E. and W.M.S.). A.S.P. was supported by a postdoctoral research fellowship award from the CFF (PIOTRO20F0), a K99/R00 Pathway to Independence award from the NIH (K99/R00 HL151806), and a Postdoc-to-Faculty Transition Award from the CFF (PIOTRO21F5). L.G.B. was supported by an NIH National Research Service Award (NRSA) training grant (T32 DK007276) and a K99/R00 Pathway to Independence award (K99/R00 HL157552). O.R. was supported by an NIH NRSA (T32 DK007276).

## Author contributions

A.S.P., L.G.B., and W.M.S. designed and conceptualized the project with input from J.S.P., A.S.P., L.G.B., and D.A.E. prepared the nanoparticle formulations. A.S.P., L.G.B., D.A.E., T.C.B., J.G., and R.M. performed the blood collection, microscopy, and biodistribution experiments. A.S.P. and D.A.E. performed the PCSK9 siRNA experiments. A.S.P., L.G.B., D.A.E., C.A., and R.M. designed and performed the antibody-mediated NP targeting experiments. A.S.P., M.E.E., and W.M.S. designed the cystic fibrosis studies and A.S.P. performed them. O.R. developed the pharmacokinetic model in consultation with A.S.P., L.G.B., and W.M.S. A.S.P. and L.G.B. wrote the initial manuscript draft. All authors contributed to reviewing and editing the manuscript.

## Competing interests

The authors declare the following competing interests: A.S.P., M.E.E. and W.M.S. are co-founders of and, at the time of preparing this manuscript, consultants for Xanadu Bio, Inc., W.M.S. is a member of the Board of Directors of Xanadu Bio, Inc., A.S.P., L.G.B., C.A., J.S.P., and W.M.S. are inventors on patent applications related to the work described here. The remaining authors declare no competing interests.
