## [Peer Review File · Nature Communications]

Reviewers' Comments:

Reviewer #1:

Remarks to the Author:

This study investigated the design strategies to enhance cell and tissue targeting of polymer nanoparticles following systemic delivery via retro-orbital injection. The strength of this study is that a large amount of blood pharmacokinetic and tissue biodistribution data for polymer nanoparticles with different physicochemical properties and varying dose levels were collected. Another strength is that the authors also create a physiologically based pharmacokinetic (PBPK) model to simulate their original data. The authors conclude that "poly(amine-co-ester) (PACE) nanoparticles and perhaps other polymer NP formulations can be designed with tunable properties to achieve desired tissue tropism for the in vivo delivery of a wide range of nucleic acid therapeutics." This study also has some weaknesses. Major and point comments are listed below.

Major comments:

1. The physiologically-based model structure (Figure 3) has a few issues. It is OK to use a cartoon style model schematic when the model structure is simple, such as for small molecules. However, for nanoparticles, the current Figure 3 is not clear. First, for a PBPK model, the blood flow direction through the lung should be from venous blood to arterial blood. This is opposite to other organs. The authors could check several other PBPK models for nanoparticles, such as this one by Li et al. 2014 (<https://pubmed.ncbi.nlm.nih.gov/24392664/>) and this one by Lin et al. 2016 (<https://pubmed.ncbi.nlm.nih.gov/25961857/>). Second, in current Figure 3, lungs, spleen, and liver have three subcompartments: intravascular space, extravascular space, and phagocytes. However, why kidneys do not have phagocytes subcompartment? Kidneys have mesangial cells. In most PBPK models for nanoparticles, liver, lungs, spleen, and kidneys have all three subcompartments. The authors can check the abovementioned models by Li et al. 2014 and Lin et al. 2016. The authors can also check more recent PBPK models for nanoparticles, such as Chen et al. 2022 (<https://www.sciencedirect.com/science/article/pii/S0168365922001171>), Chou et al. 2022 (<https://particleandfibretotoxicology.biomedcentral.com/articles/10.1186/s12989-022-00489-4>), and these two review articles by Yuan et al. 2019 (<https://www.sciencedirect.com/science/article/pii/S0022354918306695>) and Lin et al. 2022 (<https://pubs.acs.org/doi/10.1021/acsnano.2c07312>). Third, in current Figure 3, liver and lungs have all three subcompartments, intravascular space, extravascular space, and phagocytes. However, spleen only has intravascular space and phagocytes. Why does not spleen have extravascular space? This does not make sense? Fourth, Line 1051, it says "The kidney excretes NPs from the tissue compartment." In this case, where do nanoparticles go from the tissue compartment? Renal excretion has three pathways, glomerular filtration, renal reabsorption, and active secretion. For small molecules, it will go from kidney venous blood to urine. This is probably not applicable to polymeric nanoparticles because the size of the polymeric nanoparticles is apparently much bigger than the size threshold for renal excretion. So what is the mechanism of renal excretion of NPs in this model? Fifth, another big issue is the physiology of this model. PBPK models are a physiologically-based model, so the model should reflect the physiology of an organism. Right now, the model has blood, heart, lungs, liver, spleen, kidneys, and bone. However, what about other organs, such as brain, muscle, thymus, skin, etc.? Typically, in the field of PBPK modeling, other organs are represented as "Rest of body" compartment or divided as "slowly and richly perfused tissues" depending on the kinetic characteristics of the modeled substance.

2. The mathematical description of the model has an issue. Does the model consider linear kinetic mechanism only or also consider saturable kinetic mechanisms? This model considers a maximum phagocytic capacity parameter for three compartments: liver, lung, and spleen. This means that when the dose is high enough, the phagocytic cells will reach uptake saturation, thus resulting in saturable kinetic mechanism, as mentioned in the manuscript (Line 261). However, in many places of the manuscript, when explaining why their model failed to simulate a dataset, the authors mentioned that it was because their model equations are all linear (Lines 461 and 1157). This is contradictory with their model description.

3. Some of the model assumptions need additional justification. For example, Lines 1029-1030: In this case, the transport between the vascular and phagocytic sub-compartments is assumed to be

negligible, such that the modeled phagocytic transport occurs in the tissue space. Why the description of phagocytosis in the lungs is different from other tissues?

4. Some of the model equations need additional clarification. For example, Line 1054: there is only one item on the right side of the equation, which means there is a continuous flow into this subcompartment. However, it is unlikely for a substance to keep flowing in to a subcompartment, thus keep accumulating. There should be a flowing out pathway.

5. The values for some of the nanoparticle-specific parameters are not adequately justified. For example, Line 1082, what are the upper bound of the maximum uptake capacity parameters? How did the authors ensure the values of these parameters are within physiological ranges? Have the authors compared the values of their parameters with reported values from other PBPK models in the literature?

6. I do not agree with some of the result interpretation. For example, Lines 1169-1170, In general, compartmental models are linear and thus it is difficult to reproduce these dynamics mathematically without developing more specialized models. The present PBPK model is already complex enough. It already considers saturable cellular uptake. If the model fails to simulate the data, there may be some mechanisms that have not been accounted for or the model is wrong.

7. The model does not appear to be adequately validated based on Figure S2. Figure S2b result suggests that the model fails to predict concentrations in different tissues in the first three time points. In fact, results in Figure S2b suggest the performance of the model is poor.

8. It appears the model is lack of validation by an external dataset. Based on Figure S2, it appears the model was evaluated/validated by experimental data generated as part of this study. However, ideally, a PBPK model should also be validated with external/independent datasets.

9. The model evaluation criteria should be presented in the Methods section. Lines 1139-1148: the criteria of the Z values should be presented here. What Z values are considered excellent, acceptable, and poor performance/prediction? I understand the criteria are presented in one of the figure legends, but the criteria should be presented in the Methods section.

10. The uncertainty of this PBPK model should be discussed. Unlike metal nanoparticles that the concentrations of metal elements can be precisely measured using ICP-MS. In the present study, the concentrations of polymeric nanoparticles were measured using quantitative microscopy, whole organ imaging, and flow cytometry. These are all relative quantification. The authors mentioned that they had to normalized the experimental data..... (Lines 1092-1097). Thus, there is great uncertainty on the concentration data. This should be discussed.

11. I also have a concern on the reproducibility and accessibility of this model. I suggest the authors share the entire model code in the Supplementary Material or in a public repository so that other readers/researchers can access and potentially reproduce or apply their model.

12. I also have some questions about the conclusion of this study. In the abstract, it is concluded that "physiological fate can be optimized by adjusting these parameters." But how? There are not specific equations or a quantitative model for other researchers to do the optimization. There are not tangible products that other researchers can use?

13. The strengths and limitations of the present PBPK model compared to existing PBPK models for nanoparticles should be discussed.

14. The present PBPK model simulates the pharmacokinetics of the polymeric nanoparticle itself. However, for polymeric nanoparticles, one major application is to serve as a carrier to deliver active pharmaceutical ingredient, such as a small molecular drug. Upon entering into the body, the drug will be released from the polymeric nanoparticles. The pharmacokinetics of the polymeric nanoparticle itself and the pharmacokinetics of the small molecular drug are usually different. As such, when doing a PBPK model for polymeric nanoparticles, researchers also need to consider the release kinetics of the active pharmaceutical ingredient, and may have a submodel for the drug.

For the present PBPK model, even though it may simulate the kinetics of the nanoparticles, it is unclear whether it can simulate the kinetics of the active pharmaceutical ingredient.

Minor comments:

15. Lines 782-784: in Figures 4a and 4b, the PBPK-simulated concentrations are overlaid with the experimental data. However, in Figures 4c-4h, there are not experimental data? Please clarify.

16. Line 998, Table S3, why phagocytic degradation rates are only available for liver, spleen, and lung?

17. Lines 1156-1157: "Because the model equations are linear....". To some degree, the model is not exactly linear, because there are maximum uptake capacity parameters, resulting in saturable uptake.

18. Lines 606-609, Need to discuss the potential dose threshold of nanoparticles.

19. Lines 615-617: It is stated that "This finding supports the hypothesis that intentionally occupying or depleting macrophages with something other than a therapeutic NP (which may be limited in possible dose) could offer opportunities to manipulate in vivo delivery." However, this is really not a novel finding.

Reviewer #2:

Remarks to the Author:

In this manuscript, authors explored the structure-function relationship guiding physiological fate of polymeric NPs, which suggesting that polymer chemistry, vehicle characteristics, dosing and strategic co-administration of distribution modifiers might influence the blood concentration half-lives and final bio-distribution. Among all barriers, the rate of liver uptake of NPs (in particular, phagocytic uptake) is the most influential on pharmacokinetics and bio-distribution. Furthermore, two therapeutic strategies were developed for specific tissue tropism. Overall, this work is interesting and significant. But, some concerns are also needed to be solved before publication.

(1) To investigate the parameters controlling the bio-distribution of polymeric NPs, could the author explain why PACE, PACE-COOH, PACE-PEG were chose for the following experiments?

(2) In the pharmacokinetic modelling, both V_{lung} and V_{spleen} were 0.1 mL. As we know, spleen was one of the biggest organs for blood storage. More evidence and references should be provided to support authors opinions.

(3) How about the drug release profiles of different polymeric NPs encapsulating DiD? If different profiles existed in different formulations, the blood concentration and bio-distribution data in the present manuscript might not be able to represent the real situation of different polymeric NPs, indicating the whole model should be re-determined.

(4) All the bio-distribution data were obtained in healthy mice. But the situation of blood, liver, spleen, etc., varied significantly among different disease. Could the authors discuss and investigate the bio-distribution in one disease mice model, to further prove the conclusion.

(5) In Fig7d, clodronate liposomes were administrated before NPs injection to deplete macrophages for enhanced the targeted delivery of drugs. Could the authors discuss the clinic translation potential of these strategies?

Reviewer #3:

Remarks to the Author:

Hacking systemic delivery of polymer nanoparticles: designs and strategies for enhanced cell and tissue targeting

Alexandra S. Piotrowski-Daspit, Laura G. Bracaglia, David A. Eaton, Owen Richfield, Thomas C. Binns, Claire Albert, Jared Gould, Ryland D. Mortlock, Jordan S. Pober, and W. Mark Saltzman.

The authors investigated structure-function relationships guiding physiological fate of polymeric

nano particles (NPs) using high-throughput methods for measuring blood concentration and biodistribution of a library of poly (amine-co-ester) (PACE) NPs with different compositions and surface properties.

The authors used quantitative microscopy, whole organ imaging, and flow cytometry. They reported that circulation half-life as well as tissue and cell-type tropism is dependent on polymer chemistry, carrier characteristics, dosing, and strategic co-administration of distribution modifiers. They concluded that the physiological fate can be optimized by adjusting these parameters. We have identified several PACE formulations to result long half-lives and better biodistribution for therapeutic nucleic acid delivery.

The authors concluded that PACE NPs—and perhaps other polymer NP formulations—can be designed with tunable properties to achieve desired tissue reaction for the in vivo delivery of a wide range of nucleic acid therapeutics which can offer effective nucleic acid delivery vehicles for in vivo applications.

Well-written, well-prepared and well-designed investigation

My comments

1. The authors used the retro-orbital administration route, which can have some drawbacks, beside it is not that relevant in the clinical setting. The paper will benefit from a couple of experiments comparing normal IV with the retro-Orbital.

2. Introduction can be shortened.

3. The authors used IVIS for biodistribution ex-vivo- It would have been interesting from the biological point of view to study the biodistribution in vivo another labelling

Response to Reviews: Reviewer 1

- “Reviewer #1 (Remarks to the Author):

This study investigated the design strategies to enhance cell and tissue targeting of polymer nanoparticles following systemic delivery via retro-orbital injection. The strength of this study is that a large amount of blood pharmacokinetic and tissue biodistribution data for polymer nanoparticles with different physicochemical properties and varying dose levels were collected. Another strength is that the authors also create a physiologically based pharmacokinetic (PBPK) model to simulate their original data. The authors conclude that “poly(amine-co-ester) (PACE) nanoparticles and perhaps other polymer NP formulations can be designed with tunable properties to achieve desired tissue tropism for the in vivo delivery of a wide range of nucleic acid therapeutics.” This study also has some weaknesses. Major and point comments are listed below.”

We thank the reviewer for their very thoughtful and thorough review of our manuscript. Our responses to the specific points raised are provided below.

- “Major comments:
 1. *The physiologically-based model structure (Figure 3) has a few issues. It is OK to use a cartoon style model schematic when the model structure is simple, such as for small molecules. However, for nanoparticles, the current Figure 3 is not clear. First, for a PBPK model, the blood flow direction through the lung should be from venous blood to arterial blood. This is opposite to other organs. The authors could check several other PBPK models for nanoparticles, such as this one by Li et al. 2014 (<https://pubmed.ncbi.nlm.nih.gov/24392664/>) and this one by Lin et al. 2016 (<https://pubmed.ncbi.nlm.nih.gov/25961857/>).*

Thank you for this comment. We apologize for the issues with the model illustration, as well as the model it represented, and appreciate your identification of those issues. In our first model iteration—presented in the original manuscript—we added inputs to the lung and heart to represent the brachial artery and coronary artery circulation in our model. We have removed these inputs as it adds complexity that is not necessary, based on other model formulations. We have reflected these changes in the new **Figure 3**, which is reproduced below for the reviewer’s convenience:

Figure 3. Schematic of PBPK model describing PACE NP biodistribution in blood and various organs.

Second, in current Figure 3, lungs, spleen, and liver have three subcompartments: intravascular space, extravascular space, and phagocytes. However, why kidneys do not have phagocytes subcompartment? Kidneys have mesangial cells. In most PBPK models for nanoparticles, liver, lungs, spleen, and kidneys have all three subcompartments. The authors can check the abovementioned models by Li et al. 2014 and Lin et al. 2016. The authors can also check more recent PBPK models for nanoparticles, such as Chen et al. 2022 (<https://www.sciencedirect.com/science/article/pii/S0168365922001171>), Chou et al. 2022 (<https://particleandfibretoxicology.biomedcentral.com/articles/10.1186/s12989-022-00489-4>), and these two review articles by Yuan et al. 2019 (<https://www.sciencedirect.com/science/article/pii/S0022354918306695>) and Lin et al. 2022 (<https://pubs.acs.org/doi/10.1021/acsnano.2c07312>).

We agree that kidney mesangial cells may play a role in nanoparticle uptake in the kidney and that this has been considered in previous models. We have made the adjustment that the reviewer suggested. We have updated our PBPK model to incorporate phagocytic cells in the lungs, spleen and kidneys.

Third, in current Figure 3, liver and lungs have all three subcompartments, intravascular space, extravascular space, and phagocytes. However, spleen only has intravascular space and phagocytes. Why does not spleen have extravascular space? This does not make sense?

We agree and have corrected this issue, which was only in the figure and not in the model itself. Our model has always included spleen phagocytes.

Fourth, Line 1051, it says “The kidney excretes NPs from the tissue compartment.” In this case, where do nanoparticles go from the tissue compartment? Renal excretion has three pathways, glomerular filtration, renal reabsorption, and active secretion. For small molecules, it will go from kidney venous blood to urine. This is probably not applicable to polymeric nanoparticles because the size of the polymeric nanoparticles is apparently much bigger than the size threshold for renal excretion. So what is the mechanism of renal excretion of NPs in this model?

As the reviewer points out, since the polymeric nanoparticles used in this study (diameter of ~230nm) are larger than the size threshold for glomerular filtration or active secretion [1], and because we see low signal in the kidney tissue in our experiments, we assume that the small signal in the kidney is caused by sparse nanoparticle uptake by mesangial cells and that renal excretion of these nanoparticles is zero. This is further corroborated by multiple studies from our group involving polymeric NP delivery to ex vivo perfused human kidneys [2,3], in which the vast majority of untargeted NPs that remained in the kidney tissue (very few, as in the experiments conducted in the current study) were retained in glomeruli and were not filtered into the tubular space nor leaked into the interstitial space. Given that human glomerular pores are larger than that of mice, and that the PACE NPs modeled in this study are larger than the PLA-PEG NPs we have administered by perfusion to human kidneys, our assumption in the current model is that no NPs are filtered at the glomerulus. Further, the diffusion into the interstitial space is also assumed negligible in the current model, as human kidneys rejected for transplant are known to undergo massive capillary leak, and we have observed no leak into the interstitium in perfusion of human kidneys. Thus, if NPs did not diffuse out of the vascular space in these kidneys, it is unlikely that NPs would diffuse out of the vasculature in the kidneys of healthy mice.

[1] Du, B., Yu, M. & Zheng, J. Transport and interactions of nanoparticles in the kidneys. *Nat Rev Mater* **3**, 358–374 (2018), doi: 10.1038/s41578-018-0038-3

[2] Albert, C. *et al.* Monobody adapter for functional antibody display on nanoparticles for adaptable targeted delivery applications. *Nat Commun* **13**, 5998 (2022), doi:10.1038/s41467-022-33490-8

[3] Tietjen, G.T., et al., Nanoparticle targeting to the endothelium during normothermic machine perfusion of human kidneys. *Sci Transl Med*, 2017. **9**(418): p. eaam6764 (2017), doi: 10.1126/scitranslmed.aam6764

Fifth, another big issue is the physiology of this model. PBPK models are a physiologically-based model, so the model should reflect the physiology of an organism. Right now, the model has blood, heart, lungs, liver, spleen, kidneys, and bone. However, what about other organs, such as brain, muscle, thymus, skin, etc.? Typically, in the field of PBPK modeling, other organs are represented as “Rest of body” compartment or divided as “slowly and richly perfused tissues” depending on the kinetic characteristics of the modeled substance.”

We have updated the model to reflect the uptake in brain and ‘rest of body’ compartment termed “other tissues”, using parameters taken from Lin et al [1]. This is a common approach in physiologically-based models.

[1] Lin, Zhoumeng et al. A physiologically based pharmacokinetic model for polyethylene glycol-coated gold nanoparticles of different sizes in adult mice. *Nanotoxicology* **10**(2), 162-72 (2016), doi:10.3109/17435390.2015.1027314

- *“2. The mathematical description of the model has an issue. Does the model consider linear kinetic mechanism only or also consider saturable kinetic mechanisms? This model considers a maximum phagocytic capacity parameter for three compartments: liver, lung, and spleen. This means that when the dose is high enough, the phagocytic cells will reach uptake saturation, thus resulting in saturable kinetic mechanism, as mentioned in the manuscript (Line 261). However, in many places of the manuscript, when explaining why their model failed to simulate a dataset, the authors mentioned that it was because their model equations are all linear (Lines 461 and 1157). This is contradictory with their model description.”*

We are grateful for the opportunity to address this important point. We have updated our model to use the Hill Equation in modeling phagocytic uptake of these NPs, consistent with the model proposed by Lin et al. [1]. We also trained this model on blood data from early time points, to establish the correct model kinetics post-administration.

[1] Lin, Zhoumeng et al. A physiologically based pharmacokinetic model for polyethylene glycol-coated gold nanoparticles of different sizes in adult mice. *Nanotoxicology* **10**(2), 162-72 (2016), doi:10.3109/17435390.2015.1027314

- *“3. Some of the model assumptions need additional justification. For example, Lines 1029-1030: In this case, the transport between the vascular and phagocytic sub-compartments is assumed to be negligible, such that the modeled phagocytic transport occurs in the tissue space. Why the description of phagocytosis in the lungs is different from other tissues?”*

We have updated the model to reflect that phagocytes take up NPs from the vascular compartment of the lung, as opposed to the tissue compartment.

- *“4. Some of the model equations need additional clarification. For example, Line 1054: there is only one item on the right side of the equation, which means there is a continuous*

flow into this subcompartment. However, it is unlikely for a substance to keep flowing in to a subcompartment, thus keep accumulating. There should be a flowing out pathway.”

We have updated the model following the convention of Lin et al. [1] wherein phagocytes will take up and release nanoparticles, and there is passive vascular diffusion. We have little data to support clearance from the tissue compartment (by the lymph, or otherwise), thus we assume that vascular diffusion is passive and subject to the difference in NP concentration between the blood and tissue [1].

[1] Lin, Zhoumeng et al. A physiologically based pharmacokinetic model for polyethylene glycol-coated gold nanoparticles of different sizes in adult mice. *Nanotoxicology* **10**(2), 162-72 (2016), doi:10.3109/17435390.2015.1027314

- *“5. The values for some of the nanoparticle-specific parameters are not adequately justified. For example, Line 1082, what are the upper bound of the maximum uptake capacity parameters? How did the authors ensure the values of these parameters are within physiological ranges?”*

In our Monte Carlo parameter estimation scheme, all parameters are allowed to vary by orders of magnitude as specified by the parameter values initially proposed by Lin et al. Namely, we assume that the parameters used to model PACE-PEG NPs can potentially be on the same order of magnitude as the largest (smallest) value for the parameter as proposed by Lin et al. to model gold NPs. This is now described in Supplementary Materials.

We now state: “The choice of i_j was based on the bounds on each parameter used to model PEGylated gold NPs; for example, PA values for gold NPs (including 13nm and 100nm diameter NPs) were at most one order of magnitude larger or 5 orders of magnitude smaller than the values used to model the 100nm diameter gold NPs.”

Have the authors compared the values of their parameters with reported values from other PBPK models in the literature?”

By modifying the parameters from the model in Lin et al. we make the comparison between the parameter values for our PACE nanoparticles as compared to the PEGylated gold nanoparticles used in that study:

In the Results section of the manuscript titled “PACE NP PBPK Modelling”, we now state: “In comparing the parameter values for gold and PACE-PEG NPs, it is apparent that PACE-PEG NPs show substantially lower tissue uptake in comparison to gold NPs. For example, the liver’s maximum phagocytosis rate is 4 orders of magnitude lower for PACE-PEG NPs than gold NPs.”

- *“6. I do not agree with some of the result interpretation. For example, Lines 1169-1170, In general, compartmental models are linear and thus it is difficult to reproduce these dynamics mathematically without developing more specialized models. The present PBPK model is already complex enough. It already considers saturable cellular uptake.*

If the model fails to simulate the data, there may be some mechanisms that have not been accounted for or the model is wrong.”

We agree and have updated the model and its parameterization to accurately reflect the NP kinetics at early time points. Please also see our response to point 2.

- *“7. The model does not appear to be adequately validated based on Figure S2. Figure S2b result suggests that the model fails to predict concentrations in different tissues in the first three time points. In fact, results in Figure S2b suggest the performance of the model is poor.”*

We acknowledge that the model did not accurately reflect these results. In our new model, we use organ data at 3 and 48 hours along with blood data at 0.03, 1, 8, 24 and 48 hours to parameterize the model. We then use the additional data points (organ data at 6 and 24 hours and blood data at 2 and 10 hours) to validate the model.

- *“8. It appears the model is lack of validation by an external dataset. Based on Figure S2, it appears the model was evaluated/validated by experimental data generated as part of this study. However, ideally, a PBPK model should also be validated with external/independent datasets.”*

We agree that an external validation is the ideal situation for properly establishing a model for the research community. Establishing a definitive model was not our goal here; instead, we were intending to develop a model that could be refined—with other contributions from the research community—to develop hypotheses for testing in subsequent work. Further, we have not included an external validation primarily because PACE-PEG nanoparticles are (until now) unstudied in the context of nanoparticle biodistribution in mice. Since this material was originally developed in our lab and other groups have not reproduced our work, we rely on our own studies to which we may compare our results. However, none of the previous studies of PACE nanoparticle biodistribution consider the pharmacokinetics and biodistribution of PACE-PEG nanoparticles. PACE nanoparticles [1] and PACE-PEG polyplexes [2] have been studied previously, but are not comparable to the current case wherein a different material composition was used to create the nanoparticle, and new methods are used to quantify the nanoparticle concentration. We note that we are publishing the code for the model, with the hopes that other researchers who generate data using similar materials or methods will use it to generate further validations.

[1] Cui, J., et al. (2019). "Poly (amine-co-ester) nanoparticles for effective Nogo-B knockdown in the liver." *J Control Release* **304**: 259-267.

[2] Grun MK, Suberi A, Shin K, Lee T, Gomerdinger V, Moscato ZM, Piotrowski-Daspit AS, Saltzman WM. PEGylation of poly(amine-co-ester) polyplexes for tunable gene delivery. *Biomaterials*, 272:120780 (2021), doi

- *“9. The model evaluation criteria should be presented in the Methods section. Lines 1139-1148: the criteria of the Z values should be presented here. What Z values are considered excellent, acceptable, and poor performance/prediction? I understand the*

criteria are presented in one of the figure legends, but the criteria should be presented in the Methods section.”

We agree and have made this change:

In the Methods section of the manuscript titled “Mathematical Model of NP Physiologically-Based Pharmacokinetics (PBPK)”, we now state: “The typical acceptable range for PBPK model validation is a factor of two; namely, the model is considered validated if the model result is between 50% and 200% of the experimental mean. Depending on the variance of the experimental data, this convention correlates roughly to $|Z| < 20$. As shown in **Supplementary Figure 3**, the model error is mostly within this threshold.”

We are also reproducing Supplementary Figure 3 below for the reviewer’s convenience:

Figure S3. PBPK model validation. NPs in the organs are compared to model results at the time points for which the data was not used to parameterize the PBPK model of PACE-PEG biodistribution in mice, shown as the organ NP mass (relative to the liver) at 6 and 24 hours post administration (a). Mean values of the model and data are shown in blue and pink, respectively, with error bars to indicate standard deviations. (b,c) The Z value calculated based on the model and the data for the blood (b) and the corresponding model error quantified as the relative value of the model mean as compared to the data mean for the blood (c). These metrics are also shown for the organs at 6 hours post administration (d,e) and and 24 hours post administration (f,g).

- *“10. The uncertainty of this PBPK model should be discussed. Unlike metal nanoparticles that the concentrations of metal elements can be precisely measured using ICP-MS. In the present study, the concentrations of polymeric nanoparticles were measured using quantitative microscopy, whole organ imaging, and flow cytometry. These are all relative quantification. The authors mentioned that they had to normalized the experimental data..... (Lines 1092-1097). Thus, there is great uncertainty on the concentration data. This should be discussed.”*

We agree that error associated with fluorescence quantification is a challenge to overcome in modeling fluorescent nanoparticle biodistribution. We include a detailed description of how we calculate the relative distribution to non-liver organs, which relies on assumptions of cell numbers in the organs. However, this assumption still runs the risk of error. Thus, the model is not the source of error, but instead the uncertainty.

We have addressed this issue in the Discussion section of the manuscript, where we now state: *“...importantly, the results generated by the model presented here are uncertain in that the biodistribution of NPs throughout the organ compartments is calculated based on flow cytometry, which uses NP fluorescence as a means of estimating relative amounts of NPs in different organs or cell types. As a result, the biodistribution data used to parameterize our model is based on the ratio of non-liver organ fluorescence and the liver fluorescence. This naturally predisposes the model to error...”*

- *“11. I also have a concern on the reproducibility and accessibility of this model. I suggest the authors share the entire model code in the Supplementary Material or in a public repository so that other readers/researchers can access and potentially reproduce or apply their model.”*

We agree and have made our code available on GitHub, to which we reference several times throughout the revised text and the code availability statement.

- *“12. I also have some questions about the conclusion of this study. In the abstract, it is concluded that “physiological fate can be optimized by adjusting these parameters.”. But how? There are not specific equations or a quantitative model for other researchers to do the optimization. There are not tangible products that other researchers can use?”*

We appreciate the opportunity to clarify this point. In this case, we were referring to optimization in the context of experimental designs and interventions. For example, many of the decoys used are commercially available, and as such can be purchased by other researchers for use with other drug delivery vehicles. If a researcher had a NP that was accumulating in the liver macrophages but would be more impactful in liver hepatocytes, the PLGA decoy strategy we report could be employed to optimize the fate of the NPs.

We have also modified the discussion to reflect that PACE NPs are tunable, such that aspects of the polymer chemistry may be changed to alter their biodistribution and the model presented in this paper will (in the future) be applied to predict optimal PACE NP formulations for different drug delivery applications.

- “13. The strengths and limitations of the present PBPK model compared to existing PBPK models for nanoparticles should be discussed.”

We discuss the value of taking a probabilistic approach to parameterizing the model, and cite Chou et al. [1]:

In the Methods section of the manuscript titled “Mathematical Model of NP Physiologically-Based Pharmacokinetics (PBPK)”, we now state: “...subsequent model simulations can be generated by sampling these parameter distributions to generate model results that are also probabilistic in nature. As such, the model not only provides a predicted value of NP concentration in the blood and tissue compartments, but also provides the probability that the model result lies within any specified interval...”

[1] Chou WC, Cheng YH, Riviere JE, Monteiro-Riviere NA, Kreyling WG, Lin Z. Development of a multi-route physiologically based pharmacokinetic (PBPK) model for nanomaterials: a comparison between a traditional versus a new route-specific approach using gold nanoparticles in rats. *Part Fibre Toxicol* **19**(1):47 (2022), doi: 10.1186/s12989-022-00489-4.

- “14. The present PBPK model simulates the pharmacokinetics of the polymeric nanoparticle itself. However, for polymeric nanoparticles, one major application is to serve as a carrier to deliver active pharmaceutical ingredient, such as a small molecular drug. Upon entering into the body, the drug will be released from the polymeric nanoparticles. The pharmacokinetics of the polymeric nanoparticle itself and the pharmacokinetics of the small molecular drug are usually different. As such, when doing a PBPK model for polymeric nanoparticles, researchers also need to consider the release kinetics of the active pharmaceutical ingredient, and may have a submodel for the drug. For the present PBPK model, even though it may simulate the kinetics of the nanoparticles, it is unclear whether it can simulate the kinetics of the active pharmaceutical ingredient.”

We have presented data to demonstrate that PACE-PEG NPs are stable on the order of the timeline studied, such that leakage of dye and/or nucleic acid payload during this time is negligible. We have found that in the time frame of the studies described here (~48 hours), the majority of the dye remains encapsulated within the polymeric NPs. This point was addressed in an experiment described in a previous paper in which we originally described our high-throughput blood NP concentration measurements:

Bracaglia LG, Piotrowski-Daspit AS, Lin CY, Moscato ZM, Wang Y, Tietjen GT, Saltzman WM. High-throughput quantitative microscopy-based half-life measurements of intravenously injected agents. *Proc Natl Acad Sci U S A*. 2020 Feb 18;117(7):3502-3508. doi: 10.1073/pnas.1915450117. PMID: PMC7035491.

Here, we formulated NPs using a dye-conjugated polymer (PLGA-Cy5) loaded with a fluorescent dye (DiI) such that the polymer and the cargo could be imaged separately. We administered these dual-color NPs *in vitro* in HEK293 cells and intravenously in mice and collected blood samples at

various timepoints and imaged these to determine whether the labeled polymer and encapsulated dye colocalized. Further, 48 hr after NP administration at our typical experimental endpoint, we sacrificed the mice and sectioned their livers (the organ in which we typically observe the highest PLGA NP accumulation) to determine whether the fluorescent signals from the NPs and their cargo colocalized in this organ as well. In both *in vitro* and *in vivo* samples, we observed colocalization of polymer and dye signals, suggesting that using dye cargo as a tracer within the 48 hr time frame used for our high-throughput protocol provides an accurate representation of the behavior of the polymeric vehicle itself. These results are shown in **Fig. S3** of the publication referenced above describing the high-throughput blood collection method, which we have reproduced below for the reviewer's convenience:

Fig. S3. Dual-color NPs Demonstrate Colocalization of Polymeric Vehicle and Dye Cargo. Fluorescence images of dual color NPs (poly(lactic-co-glycolic acid) (PLGA)-Cy5 loaded with Dil) administered *in vitro* in HEK293 cells and *in vivo* in blood and liver samples. (a)-(d) Dil signal in (a) HEK293 cells 24 hours after treatment, (b) blood collected 2 minutes after IV NP administration, (c) blood collected 30 minutes after IV NP administration, (d) liver sections 48 hours after IV NP administration. (e)-(h) Cy5 signal in the samples described in (a)-(d). (i)-(l) Merged Dil and Cy5 signal in the samples described in (a)-(d). (m)-(p) Merged Dil and Cy5 signal in control samples. Scale bars, 100 μm.

In the current manuscript, in the Results section titled “Polymer chemistry and NP characteristics define blood concentration and biodistribution” we now state: “We administered the PACE NPs to mice intravenously (at a final blood concentration of 250 mg/mL). Blood concentration (**Figure 1b**) over time was measured from each animal using our high-throughput quantitative microscopy-based method, which requires less than 1% of the blood volume of the animal.¹⁶ A comprehensive study of tissue and cell-type tropism was performed using whole organ IVIS imaging, and flow cytometry of homogenized heart, lung, liver, spleen, kidney, and bone marrow (**Figures 1c-g**). Based on our previous studies using poly(lactic-co-glycolic acid) (PLGA) NPs, we have not observed dye leakage from solid polymeric NPs within the 48-hour time period during which blood concentration and biodistribution is assessed,¹⁶ suggesting that fluorescent signal from the dye is a reliable indicator of NP location.”

- “Minor comments:

15. Lines 782-784: in Figures 4a and 4b, the PBPK-simulated concentrations are overlaid with the experimental data. However, in Figures 4c-4h, there are not experimental data? Please clarify.”

We thank the reviewer for the opportunity to clarify this point. Due to the relative nature of the experimental data (i.e. relative fluorescence between non-liver organs and the liver), we do not have data that can serve as a direct comparison to the model output, which is in NP mass per volume organ. We have overlaid experimental data (i.e. blood NP concentrations) where available in **Figure 4**. The revised figure has been reproduced below for the reviewer’s convenience:

Figure 4. PBPK Modeling of PACE-PEG NP Biodistribution. (a) Model prediction of NP blood concentration over time with experimental datapoints overlaid. Solid lines indicates the mean of the model output, dashed lines indicate +/- SEM. Colors correspond to dosages simulated in the model, compared to corresponding data. (b) Model prediction of NP tissue concentration over time in the heart. (c) Model prediction of NP tissue concentration over time in the liver. (d) Model prediction of NP tissue concentration over time in the spleen. (e) Model prediction of NP tissue concentration over time in the kidneys. (f) Model prediction of NP tissue concentration over time in the lungs. (g) Model prediction of NP tissue concentration over time in the bone marrow. (h) Model prediction of NP tissue concentration over time in the brain and the rest of the body (combined).

- “16. Line 998, Table S3, why phagocytic degradation rates are only available for liver, spleen, and lung?”

We have added a phagocytic compartment to the kidney, as previously suggested. **Table S3** has been adjusted, as per the new model schema.

- “17. Lines 1156-1157: “Because the model equations are linear.....”. To some degree, the model is not exactly linear, because there are maximum uptake capacity parameters, resulting in saturable uptake.”

We agree. Please see responses to points 2 and 6.

- “18. Lines 606-609, Need to discuss the potential dose threshold of nanoparticles.”

We appreciate the opportunity to elaborate on this point. We have added the following sentence to this section of the Discussion section of the manuscript: “This threshold, which may be around 2.5mg/mL of polymeric NPs, is indicated by the continued increase in liver macrophages until this dose is reached, signifying they are at capacity.”

- “19. Lines 615-617: It is stated that “This finding supports the hypothesis that intentionally occupying or depleting macrophages with something other than a therapeutic NP (which may be limited in possible dose) could offer opportunities to manipulate in vivo delivery.” However, this is really not a novel finding.”

We thank the reviewer for this point, and agree that the phrasing is not written to highlight the new takeaways from this study which are more focused on the capacity limitations in macrophages for polymer NPs. We have revised the text in an effort to emphasize this.

Response to Reviews: Reviewer 2

Reviewer #2 (Remarks to the Author):

- *“In this manuscript, authors explored the structure-function relationship guiding physiological fate of polymeric NPs, which suggesting that polymer chemistry, vehicle characteristics, dosing and strategic co-administration of distribution modifiers might influence the blood concentration half-lives and final bio-distribution. Among all barriers, the rate of liver uptake of NPs (in particular, phagocytic uptake) is the most influential on pharmacokinetics and bio-distribution. Furthermore, two therapeutic strategies were developed for specific tissue tropism. Overall, this work is interesting and significant. But, some concerns are also needed to be solved before publication.”*

We thank the reviewer for their encouraging remarks about our manuscript. We also appreciate the opportunity to further improve our work in response to the concerns raised. Our responses to the individual points raised by Reviewer #2 are provided below:

- *“(1)To investigate the parameters controlling the bio-distribution of polymeric NPs, could the author explain why PACE, PACE-COOH, PACE-PEG were chose for the following experiments?”*

We thank the reviewer for the opportunity to clarify this point. These families of PACE polymers (PACE, PACE-COOH, and PACE-PEG) have recently been developed in our laboratory and described in-depth in one of our previous publications:

Kauffman AC, Piotrowski-Daspit AS, Nakazawa KH, Jiang Y, Datye A, Saltzman WM. Tunability of Biodegradable Poly(amine- co-ester) Polymers for Customized Nucleic Acid Delivery and Other Biomedical Applications. *Biomacromolecules*. 19(9):3861-3873 (2018), doi: 10.1021/acs.biomac.8b00997.

These polymers are well characterized in terms of chemical and physical properties. Each of these materials incorporates elements to improve nucleic acid delivery in recent publications, though their behavior *in vivo* has not been thoroughly studied before, motivating the current study.

In the Results section titled “Polymer chemistry and NP characteristics define blood concentration and biodistribution” we now state: “Taking advantage of our PACE library of materials (Figure 1a), we prepared NP formulations with various PACE polymer chemistries: PACE (an unmodified PACE polymer), PACE-COOH, and PACE-PEG. We chose these materials as they have been previously well-characterized with demonstrated efficient nucleic acid loading and delivery.¹³”

- *“(2)In the pharmacokinetic modelling, both V_{lung} and V_{spleen} were 0.1 mL. As we know, spleen was one of the biggest organs for blood storage. More evidence and references should be provided to support authors opinions.”*

We have updated the physiological parameters of our model in line with Lin et al. [1], which further states that V_{Lung} and V_{Spleen} are equal to 0.1ml, from a different reference of physiological values [2] than the reference used originally [3].

[[1] Lin, Zhoumeng et al. A physiologically based pharmacokinetic model for polyethylene glycol-coated gold nanoparticles of different sizes in adult mice. *Nanotoxicology* **10**(2), 162-72 (2016), doi:10.3109/17435390.2015.1027314

[2] Davies, B. and T. Morris. Physiological parameters in laboratory animals and humans. *Pharm Res* **10**(7): 1093-1095 (1993), doi: 10.1023/a:1018943613122

[3] Brown RP, Delp MD, Lindstedt SL, Rhomberg LR, Beliles RP. Physiological parameter values for physiologically based pharmacokinetic models. *Toxicol Ind Health* **13**:407-84 (1997), doi: 10.1177/074823379701300401

- “(3)How about the drug release profiles of different polymeric NPs encapsulating DiI? If different profiles existed in different formulations, the blood concentration and bio-distribution data in the present manuscript might not be able to represent the real situation of different polymeric NPs, indicating the whole model should be re-determined.”

This is an important point, which we have considered. We have found that in the time frame of the studies described here (~48 hours), the majority of the dye remains encapsulated within the polymeric NPs. This point was addressed in an experiment described in a previous paper in which we originally described our high-throughput blood NP concentration measurements:

Bracaglia LG, Piotrowski-Daspit AS, Lin CY, Moscato ZM, Wang Y, Tietjen GT, Saltzman WM. High-throughput quantitative microscopy-based half-life measurements of intravenously injected agents. *Proc Natl Acad Sci U S A*. 2020 Feb 18;117(7):3502-3508. doi: 10.1073/pnas.1915450117. PMID: PMC7035491.

Here, we formulated NPs using a dye-conjugated polymer (PLGA-Cy5) loaded with a fluorescent dye (DiI) such that the polymer and the cargo could be imaged separately. We administered these dual-color NPs *in vitro* in HEK293 cells and intravenously in mice and collected blood samples at various timepoints and imaged these to determine whether the labeled polymer and encapsulated dye colocalized. Further, 48 hr after NP administration at our typical experimental endpoint, we sacrificed the mice and sectioned their livers (the organ in which we typically observe the highest PLGA NP accumulation) to determine whether the fluorescent signals from the NPs and their cargo colocalized in this organ as well. In both *in vitro* and *in vivo* samples, we observed colocalization of polymer and dye signals, suggesting that using dye cargo as a tracer within the 48 hr time frame used for our high-throughput protocol provides an accurate representation of the behavior of the polymeric vehicle itself. These results are shown in **Fig. S3** of the publication referenced above describing the high-throughput blood collection method, which we have reproduced below for the reviewer's convenience:

Fig. S3. Dual-color NPs Demonstrate Colocalization of Polymeric Vehicle and Dye Cargo. Fluorescence images of dual color NPs (poly(lactic-co-glycolic acid) (PLGA)-Cy5 loaded with Dil) administered *in vitro* in HEK293 cells and *in vivo* in blood and liver samples. (a)-(d) Dil signal in (a) HEK293 cells 24 hours after treatment, (b) blood collected 2 minutes after IV NP administration, (c) blood collected 30 minutes after IV NP administration, (d) liver sections 48 hours after IV NP administration. (e)-(h) Cy5 signal in the samples described in (a)-(d). (i)-(l) Merged Dil and Cy5 signal in the samples described in (a)-(d). (m)-(p) Merged Dil and Cy5 signal in control samples. Scale bars, 100 μm .

In the current manuscript, in the Results section titled “Polymer chemistry and NP characteristics define blood concentration and biodistribution” we now state: “We administered the PACE NPs to mice intravenously (at a final blood concentration of 250 mg/mL). Blood concentration (**Figure 1b**) over time was measured from each animal using our high-throughput quantitative microscopy-based method, which requires less than 1% of the blood volume of the animal.¹⁶ A comprehensive study of tissue and cell-type tropism was performed using whole organ IVIS imaging, and flow cytometry of homogenized heart, lung, liver, spleen, kidney, and bone marrow (**Figures 1c-g**). Based on our previous studies using poly(lactic-co-glycolic acid) (PLGA) NPs, we have not observed dye leakage from solid polymeric NPs within the 48-hour time period during which blood concentration and biodistribution is assessed,¹⁶ suggesting that fluorescent signal from the dye is a reliable indicator of NP location.”

- “(4)All the bio-distribution data were obtained in healthy mice. But the situation of blood, liver, spleen, etc., varied significantly among different disease. Could the authors discuss and investigate the bio-distribution in one disease mice model, to further prove the conclusion.”

We thank the reviewer for bringing up this point and certainly agree that disease state is likely to influence biodistribution and circulation half-life. It will be important to consider this for each potential therapeutic application. While it is not possible to extensively test how our PACE NPs perform in multiple disease animal models, we have collected data in a mouse model of cystic fibrosis (CF), one of the disease targets our laboratory is interested in. We have also recently published on the biodistribution of PLGA NPs in this F508del CF model:

Piotrowski-Daspit AS, Barone C, Lin CY, Deng Y, Wu D, Binns TC, Xu E, Ricciardi AS, Putman R, Garrison A, Nguyen R, Gupta A, Fan R, Glazer PM, Saltzman WM, Egan ME. In vivo correction of cystic fibrosis mediated by PNA nanoparticles. *Sci Adv.* 2022 Oct 7;8(40):eabo0522. doi: 10.1126/sciadv.abo0522. Epub 2022 Oct 5. PMID: 36197984; PMCID: PMC9534507.

To address the reviewer’s question, we measured the PACE NP biodistribution data in the same disease model. That data is now included in **Supplementary Figure 2**, which has been reproduced below for the reviewer’s convenience:

Figure S2. PACE and PACE-PEG NP biodistribution in the F508del CF mouse model. (a) Schematic of the F508del CF mouse model illustrating the CF-associated 3 bp deletion in the CFTR gene. (b) Representative end-point IVIS analysis of PACE and PACE-PEG NP uptake in various organs (heart, lungs, pancreas, liver, spleen, kidneys, bone, and gastrointestinal (GI) tract). End-point analyses of (c) whole organ fluorescence quantification of PACE and PACE-PEG NP uptake in various organs (n = 3 mice per group per organ; error bars represent SEM), (d) %NP+ cells in homogenized organs by flow cytometry (n = 3 mice per group per organ; error bars represent SEM).

In the Results section titled “Polymer chemistry and NP characteristics define blood concentration and biodistribution” we now state: “Beyond NP characteristics, we have also found that disease state/pathophysiology can impact biodistribution. We compared the biodistribution of a subset of PACE NPs in a cystic fibrosis (CF) mouse model harboring the F508del CF mutation, and while the general trends are the same in that PACE-PEG NPs accumulate more broadly than unPEGylated NPs, PACE Classic NPs also accumulated in the lungs of CF animals in contrast to our observations in wild-type animals, where no lung accumulation was observed (Supplementary Figure 2).”

Further, in the Discussion section we now state: “Disease state is also important to consider, as pathophysiology can impact NP trafficking.”

- “(5) In Fig 7d, clodronate liposomes were administered before NPs injection to deplete macrophages for enhanced the targeted delivery of drugs. Could the authors discuss the clinic translation potential of these strategies?”

We appreciate the opportunity to expand on the clinical potential of decoy usage. In our study, clodronate liposomes were a useful tool to demonstrate that depletion of phagocytic cells like macrophages could enhance or broaden the biodistribution of PACE NPs (including NPs with antibodies conjugated to the surface, as in Figure 7) as a proof of concept. However, we did not mean to suggest that clodronate liposomes should be translated to the clinic. Macrophage depletion would likely be a safety concern. Instead, we believe temporary “occupation” of macrophages by more biocompatible alternatives is possible. This is why we also wanted to demonstrate the utility of the other decoys described, such as intralipid and PLGA NPs in Figure 5, as we believe these have more potential for translation. Intralipid, for example, is already used as an intravenous infusion in the clinic, and PLGA is a major component of several FDA-approved products. Indeed, PLGA NPs were effective in Figure 6 when used as a decoy to boost PCSK9 knockdown mediated by siRNA in PACE NPs.

In the Discussion section of the manuscript we now state: “Regarding the potential for clinical translation, the safety of decoy formulations in addition to therapeutic PACE formulations will need to be assessed. Temporary phagocytic occupation with biocompatible materials as opposed to depletion of phagocytes is the most likely strategy to succeed in the clinic. Both intralipid and PLGA NPs already have favorable safety profiles, as intralipid is already used in the clinic and PLGA particles are part of several FDA-approved products for drug delivery.”

Response to Reviews: Reviewer 3

Reviewer #3 (Remarks to the Author):

- *“The authors investigated structure-function relationships guiding physiological fate of polymeric nano particles (NPs) using high-throughput methods for measuring blood concentration and biodistribution of a library of poly (amine-co-ester) (PACE) NPs with different compositions and surface properties. The authors used quantitative microscopy, whole organ imaging, and flow cytometry. They reported that circulation half-life as well as tissue and cell-type tropism is dependent on polymer chemistry, carrier characteristics, dosing, and strategic co-administration of distribution modifiers. They concluded that the physiological fate can be optimized by adjusting these parameters. We have identified several PACE formulations to result long half-lives and better biodistribution for therapeutic nucleic acid delivery. The authors concluded that PACE NPs—and perhaps other polymer NP formulations—can be designed with tunable properties to achieve desired tissue reaction for the in vivo delivery of a wide range of nucleic acid therapeutics which can offer effective nucleic acid delivery vehicles for in vivo applications. Well-written, well-prepared and well-designed investigation”*

We very much appreciate these encouraging remarks! Our responses to the additional comments from this reviewer are provided below:

- *“My comments
1. The authors used the retro-orbital administration route, which can have some drawbacks, beside it is not that relevant in the clinical setting. The paper will benefit from a couple of experiments comparing normal IV with the retro-Orbital.”*

We thank the reviewer for noting this potential cause for differences in nanoparticle biodistribution. In a recent publication focused on developing the high-throughput methods used in the current study, we studied the biodistribution of dye-loaded polymeric PLGA NPs after tail vein or retro-orbital IV administration. We observed no difference in blood concentration profiles, whole organ fluorescence, or flow cytometry results, suggesting that biodistribution is comparable following retro-orbital or tail vein administration of the same NP type. This data is shown in **Figure 2** of our recent publication and is reproduced below for the reviewer’s convenience:

Fig. 2. Tail-Vein and Retro-Orbital Injections of Polymeric Nanoparticles Yield Identical Measurements of Half-life. (a) Schematic of polymeric nanoparticles composed of poly(lactic acid)-poly(ethylene glycol) copolymer (PLA-PEG) encapsulating fluorescent dye and routes of IV administration. (b) Representative fluorescence images of dye-loaded nanoparticles at different concentrations in blood. (c) Circulation half-life measurements after IV administration of fluorescent polymeric nanoparticles using quantitative microscopy. Error bars represent standard error of the mean. End-point (d,e) flow cytometry and (f) in vivo imaging system (IVIS) analyses of NP uptake in various tissues. Scale bars, 100 μm .

In the current manuscript we have noted this point for additional clarification. In the Methods section titled “*In vivo* mouse NP administration and mouse blood collection”, we now state: “We have previously demonstrated that biodistribution following retro-orbital injection is equivalent to biodistribution following tail vein injection.”¹⁶”

- “2. Introduction can be shortened.”

We thank the reviewer for this suggestion to improve our manuscript. In the revised text, we have made an effort to make the introduction more concise, reducing it by ~70 words. The revised introduction is reproduced below for the reviewer’s convenience:

“Targeted drug delivery *in vivo* is a complex challenge, with multiple barriers at the levels of the organism, organs/tissues, and individual cells.⁴ Non-viral lipid- or polymer-based vehicles can improve delivery efficiency of therapeutic molecules. In the context of nucleic acid delivery, while non-viral vehicles can be effective at cargo encapsulation, protection of cargo from nuclease degradation, and transport across the cell membrane, the primary challenge is still sufficient accumulation in target tissues instead of non-target tissues, where they can lead to side-effects. Standard therapeutic development pipelines, in which candidate vehicles are identified in cell culture studies, are often not predictive of what happens when these agents are administered to an animal. For example, we and others have found differences in vehicle characteristics affecting cell uptake and release of cargo following *in vitro* vs. *in vivo* delivery.⁵⁻⁷ There are good reasons for this: traditional cell culture models are simplified and often contain one cell type without normal tissue architecture, fluid flows, and other complexities found *in vivo*. Several studies have highlighted the need to study delivery in more physiologically relevant contexts, including high-throughput screens *in vivo*.⁸

The majority of studies on polymeric vehicles encapsulating nucleic acids employ local delivery unless hepatic tissue is the disease target.⁴ Local or compartmental administration methods—such as intranasal or intratracheal instillation to the lungs or intraocular administration—can circumvent systemic clearance mechanisms. However, local delivery is not suitable for the treatment of *in situ* tumors, internal organs, and diseases that affect multiple organ systems. Moreover, it may not provide access to the disease-associated cell types within a tissue. Intravenous (IV) administration can theoretically provide opportunities for delivery vehicles to reach almost any tissue in the body. But there are barriers; the single most formidable barrier to IV delivery of particulate carriers is clearance by the mononuclear phagocyte system (MPS) (sometimes also known as the reticuloendothelial system (RES))— i.e. vehicle uptake by intravascular phagocytic cells present primarily in the liver and spleen.⁹⁻¹² Clearance by these phagocytes reduces the dose of vehicles available in circulation and limits possible accumulation in intended tissues. To overcome this barrier, vehicles can be designed with strong affinity to a particular tissue (tissue tropism) so that they can accumulate in that tissue faster than they are cleared by phagocytes. Other carriers with abilities to evade phagocyte clearance or extend circulation time allow for more chances to accumulate in areas of interest. Understanding the characteristics that define the behavior of delivery vehicles *in vivo* is important for future advances in delivery vehicle design.

We have designed a library of polymeric delivery vehicles as well as high-throughput tools to study the structure-function relationships guiding the physiological fate of nanomedicines. Poly(amine-co-esters) (PACEs) are a family of tunable, biodegradable, and biocompatible polymers designed for nucleic acid delivery (**Figure 1**). We have used these materials to deliver a wide range of nucleic acids, such as siRNA, mRNA, and pDNA, both in cell culture and *in vivo*.¹³⁻¹⁵ The ability to control the physical and chemical properties of PACEs—as well as the size and surface properties of the resulting vehicles including the incorporation of targeting ligands—makes this platform ideal for studying the effects of polymer chemistry and nanoparticle (NP) characteristics on physiological fate *in vivo* after IV delivery. We hypothesize that these factors affect both circulation time and tissue tropism. We can also use PACE vehicles to determine the effects of dosing⁹ and strategic co-administration of multiple formulations on NP clearance rate. To assess these variables, we employ a recently developed high-throughput tool to measure NP blood concentration¹⁶ combined with thorough biodistribution assessment using whole organ imaging

and flow cytometry. We then use these comprehensive data sets to train a pharmacokinetic model describing PACE NP circulation and biodistribution. This combined approach—using multiple experimental measurements combined with physiology-based computer models—enables us to gain a comprehensive understanding of delivery vehicle fate, and to determine the factors that influence delivery.

Overall, we have identified several parameters that are key for controlling the biodistribution of polymeric NPs. NP properties such as size, charge, and morphology play an important role, but are often constrained by the type of cargo one intends to deliver. Our multimodal experiments and computational modeling establish that dosing, co-administration of “decoys” to limit phagocytic clearance, and antibody-mediated targeting can impact the physiological fate and therapeutic effects of systemic NP delivery.”

- “3. The authors used IVIS for biodistribution ex-vivo- It would have been interesting from the biological point of view to study the biodistribution in vivo another labelling”

We thank the reviewer for bringing up this point. In our experience, *in vivo* IVIS imaging for biodistribution assessment is not as clear as *ex vivo* IVIS imaging of organs in terms of the origin of fluorescent signal, which is why we did not choose to include this type of data in the manuscript. We have also observed autofluorescence from an animal’s hair, for example. From what we can tell, however, results are consistent with our *ex vivo* assessments (ex. PACE-PEG NPs have more widespread distribution). We provide an example below of unpublished data comparing various PACE NP formulations described in our manuscript as well as PLGA NPs:

untreated control PLGA PACE 60% PDL Classic PACE 60% PDL COOH PACE 60% PDL PEG Classic

All of the IVIS data provided in the manuscript was captured immediately after euthanasia and organ harvest, so as to best represent the most recent *in vivo* state of the animals.

In the Methods section of the manuscript titled “Biodistribution end-point analyses”, we now state: “Organs (heart, lungs, liver, spleen, kidneys, and bone) were harvested. PACE NP accumulation in the organs was visualized and quantified **immediately after harvest** using an In Vivo Imaging System (IVIS) (Perkin Elmer).”

Reviewers' Comments:

Reviewer #2:

Remarks to the Author:

[Note from the Editor: Reviewer #2 was asked to also review the response given to the original Reviewer #3.]

Thank authors for their careful and detailed responses. All my concerns have been addressed point by point.

As requested by the editor, according to the responses for reviewer 3, I consider that the authors have addressed the comments within the limits of their ability. In particular, some pre-experiment results were displayed to address the reviewer's concern.

Reviewer #4:

Remarks to the Author:

[Note from the Editor: Reviewer #4 was asked to review the response given to the original Reviewer #1.]

This article investigated the relationship between nanoparticle structure and functions of PACE-NPs. Using multiple experimental methods and the computational method of PBPK modeling, the authors successfully quantified the blood pharmacokinetic profile of the testing NPs and evaluated the changes of main organ biodistributions. The study is good for the advantages of multiple measures of blood concentrations which describe the blood pharmacokinetic profile well. The authors made a conclusion that PACE NPs and other polymer NP formulations can be designed with tunable properties to achieve desired tissue tropism for in vivo delivery. Generally, the study is well-designed and the article is well-written. However, This article still has some limitations to be addressed before publication.

1. In the Line 266, the word of "body" is not very accurate. In Figure 3, the rest of body was named as "other tissues". Therefore, the authors need to define these terms.
2. The description of the compartment of the brain is not clear. Is the brain listed as a separate compartment? Based on Table S3, it seems that there is a set of separate parameters for the brain, which indicate a separate compartment. If so, why is it not included in Figure 3, the schematic of the PBPK model? If not, I think "the rest of body" can include all other tissues besides the six target organs and the brain does not need to be listed separately. If this compartment is necessary, I would suggest adding more justification to the main text.
3. Line 293-294, a citation is needed here.
4. The statistics of z scores to evaluate model accuracy are not sufficient. Usually, a R2 (and a RMSE) is used for the goodness of fit for model validation. Based on Figure 4, the modeling of the blood data does not fit very well, which may result in different accumulation patterns in the liver, heart vs other organs. In the liver and heart, it did not reach the peak concentrations within 48 hours, while the Cmax in the spleen, lung and bone generally appeared at 8 hours. Is this phenomenon consistent with the experimental results? When did the peak concentrations appear in the liver and heart in the experiment?
5. In Figure 4A, the experimental data for different doses generally showed a short period of accumulation (concentrations increasing) in the first few hours, while the simulated data showed a gradual decline as the author said in Line 478. This phenomenon was also shown in the authors' previous work (Ref.16) discussing the pharmacokinetic profile difference between retro-orbital injection and intravenous injection. This difference is one of the reasons for less goodness of fit. I agree with the authors that this model is still usable in this study. However, this point should be discussed in this manuscript for the model selection and model applicability in this study.
6. Another issue of Figure 4 is the authors only validated the model using the blood data but no organ biodistribution data, which will cause the unreliability of Figures 4b-h. Moreover, due to no validation with experimental tissue concentrations, it is difficult to examine the parameters for tissues. Therefore, I don't think Figures 4b-h should be included. In contrast, the authors said "The biodistribution data used to parameterize our model is based on the ratio of non-liver organ fluorescence and the liver fluorescence" in Line 654. This comparison between the simulation

results and the experimental results (the ratio of non-liver organs and the liver fluorescence) should be presented.

7. The code is not available at the given link (https://github.com/omrichfield/PACE-PBPK-Monte-Carlo_public)

Response to Reviews: Reviewer #2

- “Reviewer #2 (Remarks to the Author):

[Note from the Editor: Reviewer #2 was asked to also review the response given to the original Reviewer #3.]

Thank authors for their careful and detailed responses. All my concerns have been addressed point by point.

As requested by the editor, according to the responses for reviewer 3, I consider that the authors have addressed the comments within the limits of their ability. In particular, some pre-experiment results were displayed to address the reviewer's concern.”

We thank the reviewer again for their thoughtful suggestions to improve our manuscript, and for taking the time to review our responses to concerns raised by Reviewer #3.

Response to Reviews: Reviewer #4

- “Reviewer #4 (Remarks to the Author):

[Note from the Editor: Reviewer #4 was asked to review the response given to the original Reviewer #1.]

This article investigated the relationship between nanoparticle structure and functions of PACE-NPs. Using multiple experimental methods and the computational method of PBPK modeling, the authors successfully quantified the blood pharmacokinetic profile of the testing NPs and evaluated the changes of main organ biodistributions. The study is good for the advantages of multiple measures of blood concentrations which describe the blood pharmacokinetic profile well. The authors made a conclusion that PACE NPs and other polymer NP formulations can be designed with tunable properties to achieve desired tissue tropism for in vivo delivery. Generally, the study is well-designed and the article is well-written. However, This article still has some limitations to be addressed before publication.”

We thank the new reviewer for their positive overall impression of our manuscript and for the additional comments from their thorough review. We also appreciate Reviewer #4 taking the time to review our response to the comments from Reviewer #1. Our responses to the new specific points raised are provided below.

- “1. In the Line 266, the word of “body” is not very accurate. In Figure 3, the rest of body was named as “other tissues”. Therefore, the authors need to define these terms.”

We agree with the reviewer on this point and have eliminated the use of “body” as a compartment name. We now define the “other tissues” compartment as all tissues separate from the six measured compartments and the brain compartment, which we have added to **Figure 3**. The revised **Figure 3** is reproduced below for the reviewer’s convenience:

Figure 3. Schematic of PBPK model describing PACE NP biodistribution in blood and various organs. The “other tissues” compartment is defined as all tissues separate from the ones displayed here.

- “2. The description of the compartment of the brain is not clear. Is the brain listed as a separate compartment? Based on Table S3, it seems that there is a set of separate parameters for the brain, which indicate a separate compartment. If so, why is it not included in Figure 3, the schematic of the PBPK model? If not, I think “the rest of body” can include all other tissues besides the six target organs and the brain does not need to be listed separately. If this compartment is necessary, I would suggest adding more justification to the main text.”

We thank the reviewer for the opportunity to clarify this point. We have included the brain as a separate compartment in **Figure 3** (reproduced above). We included a brain compartment so as not to make unnecessary changes to the model from which we derived the model used in our study, which included a brain compartment. We kept the brain compartment in our model even though

PACE NP uptake in the brain has been shown to be minimal, which is reported in the following publication (and cited in the manuscript):

Cui, J. *et al.* Poly(amine-co-ester) nanoparticles for effective Nogo-B knockdown in the liver. *J Control Release* **304**, 259-267 (2019). [https://doi.org:10.1016/j.jconrel.2019.04.044](https://doi.org/10.1016/j.jconrel.2019.04.044)

We also include here an example of our unpublished internal data comparing distribution of a PACE formulation to untreated control and poly(lactic-co-glycolic) acid (PLGA), again with no signal in the brain:

Thus, when we portray NP uptake in compartments other than the six measured compartments, we lump the brain and ‘other tissues’ compartment results together in **Figure 4h**.

We discuss the inclusion of the brain compartment in lines 267-273 of the revised manuscript, where we now state: “We have previously shown that PACE NPs do not accumulate significantly in the brain, thus the brain tissue is not included in our experimental results. Nevertheless, the previously published model from which we derived the model for this study included a brain compartment with associated permeability to NPs, thus we have kept this compartment with associated NP absorption. Uptake of NPs in the brain is combined with the ‘other tissues’ compartment in **Figure 4.**”

- “3. Line 293-294, a citation is needed here.”

We agree and have included the relevant citation. We now state: “As stated previously, the model used in our study was originally developed using biodistribution data from PEGylated gold NPs in mice.¹⁸”

Lin, Z., Monteiro-Riviere, N. A. & Riviere, J. E. A physiologically based pharmacokinetic model for polyethylene glycol-coated gold nanoparticles of different sizes in adult mice. *Nanotoxicology* **10**, 162-172 (2016). <https://doi.org/10.3109/17435390.2015.1027314>

- “4. The statistics of z scores to evaluate model accuracy are not sufficient. Usually, a R² (and a RMSE) is used for the goodness of fit for model validation.”

We agree and have included a calculation of R² and RMSE to evaluate model fit to the blood data as well as the relative biodistribution of NPs to the non-liver tissues. These calculations are now included in **Supplementary Figure 3**, which has been reproduced below for the reviewer’s convenience:

Figure S3. PBPK model validation. NPs in the organs are compared to model results at the time points for which the data was not used to parameterize the PBPK model of PACE-PEG biodistribution in mice, shown as the organ NP mass (relative to the liver) at 6 and 24 hours post administration (**a**). Mean values of the model and data are shown in blue and pink, respectively, with error bars to indicate standard deviations. (**b,c**) The Z value calculated based on the model and the data for the blood (**b**) and the corresponding model error quantified as the relative value of the model mean as compared to the data mean for the blood (**c**). These metrics are also shown for the organs at 6 hours post administration (**d,e**) and and 24 hours post administration (**f,g**). (**h**) Actual and model-predicted PACE-PEG NP blood concentration at all time points for all doses are compared, with calculated R^2 and root mean squared error (RMSE). (**i**) Organ mass of NPs

normalized to the liver at all time points (3, 6, 24 and 48 hours post-administration) are compared, with calculated R^2 and RMSE.

- *“Based on Figure 4, the modeling of the blood data does not fit very well, which may result in different accumulation patterns in the liver, heart vs other organs. In the liver and heart, it did not reach the peak concentrations within 48 hours, while the C_{max} in the spleen, lung and bone generally appeared at 8 hours. Is this phenomenon consistent with the experimental results? When did the peak concentrations appear in the liver and heart in the experiment?”*

We thank the reviewer for this comment and the opportunity to clarify. Because of the relative nature of the data used to parameterize the model, the kinetics of NP uptake in the organs is difficult to ascertain; according to **Supplementary Figure 1**, the spleen and liver show maximum uptake at 24 hours, while the remaining organs appear to show maximum uptake at 48 hours. However, when normalizing the NP fluorescence in each organ by the liver, these kinetics are altered, as reflected by the model results in **Figure 4**. Namely, the model suggests that uptake is maximized in the non-liver organs at approximately 8 hours, when the uptake in the liver is low. The exception is the heart, which has such low uptake that its pharmacokinetic profile is dominated by the measurements from the liver. As a result, the heart and liver both show a steady increase in NP accumulation over 48 hours post-administration. While this profile does not appear to fit the profile of the liver as per **Supplementary Figure 1**, the requirement that the liver be used as a means of normalizing NP fluorescence in non-liver organs prohibits using individual liver time points to normalize the liver’s overall time course which would then require that the non-liver organs adhere to this same rule, which is contradictory to the time course generated by normalizing these organs’ concentration to the liver at each time point.

- *“5. In Figure 4A, the experimental data for different doses generally showed a short period of accumulation (concentrations increasing) in the first few hours, while the simulated data showed a gradual decline as the author said in Line 478. This phenomenon was also shown in the authors’ previous work (Ref.16) discussing the pharmacokinetic profile difference between retro-orbital injection and intravenous injection. This difference is one of the reasons for less goodness of fit. I agree with the authors that this model is still usable in this study. However, this point should be discussed in this manuscript for the model selection and model applicability in this study.”*

We thank the reviewer for this comment.

We reference this model limitation in lines 677-681 of the revised manuscript, where we now state: “...the model used in this study mainly provides a means of hypothesis generation and must be further improved and tested in subsequent studies. Importantly, future iterations of the model will investigate the mechanism of accumulation of NPs in the blood within the first 8 hours post-administration, which is the primary source of error in the predicted blood pharmacokinetic profile.”

- *“6. Another issue of Figure 4 is the authors only validated the model using the blood data but no organ biodistribution data, which will cause the unreliability of Figures 4b-h. Moreover, due to no validation with experimental tissue concentrations, it is difficult to examine the parameters for tissues. Therefore, I don’t think Figures 4b-h should be included. In contrast, the authors said “The biodistribution data used to parameterize our model is based on the ratio of non-liver organ fluorescence and the liver fluorescence” in Line 654. This comparison between the simulation results and the experimental results (the ratio of non-liver organs and the liver fluorescence) should be presented.”*

We apologize if this was unclear, but we did indeed previously include a thorough model validation in **Supplementary Figure 3** (reproduced above), which includes the model results for all non-liver tissues normalized to the liver and compares to the associated experimental data, both in calculating Z scores as well as the ratio of the data to the model prediction. In this revision, we have included in this figure a calculation of R^2 and the RMSE for both blood NP concentration and non-liver organ biodistribution.

- *“7. The code is not available at the given link (https://github.com/omrichfield/PACE-PBPK-Monte-Carlo_public)”*

We apologize for the inconvenience and have verified that the entirety of the model code is available at this link.

- *“Reviewer #4 (Remarks on code availability):*

The code is not available using this link”

Please see our response to point 7, above.

Reviewers' Comments:

Reviewer #4:

Remarks to the Author:

The authors have addressed all my questions. I believe it is suitable to publish it.

Response to Reviews: Reviewer #4

- “Reviewer #4 (Remarks to the Author):

The authors have addressed all my questions. I believe it is suitable to publish it.”

We thank the new reviewer again for their time. We believe their comments have improved this manuscript.

- “Reviewer #4 (Remarks on code availability):

The code is useful to reproduce the results of the paper. The README file is not very sufficient. But I still believe it is clear to run the code for other researchers.”

We agree that the README file required updates and expansion. We have addressed the main aspects of running the code, and have explained the function of each R script. These additions are included in the new README file on Github.